# STIC2 selectively binds ribosome-nascent chain complexes in the cotranslational sorting of *Arabidopsis* thylakoid proteins

Dominique S Stolle [1,7], Lena Osterhoff[1,7], Paul Treimer[1], Jan Lambertz[2], Marie Karstens[1], Jakob-Maximilian Keller [3], Ines Gerlach [3], Annika Bischoff[1], Beatrix Dünschede[1], Anja Rödiger[2], Christian Herrmann [4], Sacha Baginsky[2], Eckhard Hofmann[5], Reimo Zoschke[3], Ute Armbruster [3,6], Marc M Nowaczyk[2] & Danja Schünemann [1✉]

## Abstract

**Chloroplast-encoded multi-span thylakoid membrane proteins are crucial for photosynthetic complexes, yet the coordination of their biogenesis remains poorly understood. To identify factors that specifically support the cotranslational biogenesis of the reaction center protein D1 of photosystem (PS) II, we generated and affinity-purified stalled ribosome-nascent chain complexes (RNCs) bearing D1 nascent chains. Stalled RNCs translating the soluble ribosomal subunit uS2c were used for comparison. Quantitative tandem-mass spectrometry of the purified RNCs identified around 140 proteins specifically associated with D1 RNCs, mainly involved in protein and cofactor biogenesis, including chlorophyll biosynthesis, and other metabolic pathways. Functional analysis of STIC2, a newly identified D1 RNC interactor, revealed its cooperation with chloroplast protein SRP54 in the de novo biogenesis and repair of D1, and potentially other cotranslationally-targeted reaction center subunits of PSII and PSI. The primary binding interface between STIC2 and the thylakoid insertase Alb3 and its homolog Alb4 was mapped to STIC2's β-sheet region, and the conserved Motif III in the C-terminal regions of Alb3/4.**

**Keywords** *Arabidopsis*; Cotranslational Targeting; Photosystem II; STIC2; Thylakoid Membrane
**Subject Categories** Organelles; Plant Biology

## Introduction

The biogenesis and maintenance of the photosynthetically active thylakoid membrane of chloroplasts requires the cotranslational targeting of plastid-encoded thylakoid membrane proteins, which mediate electron transport and ATP synthesis during photosynthesis (Jagendorf and Michaels, 1990; Kim et al, 1991; Zhang et al, 1999; Zoschke and Barkan, 2015). The synthesis of these proteins begins at soluble ribosomes that can be indirectly tethered to the thylakoid membrane by mRNA-associated factors. With the emergence of the first transmembrane helix (TMH) of the nascent peptide, a tight nuclease-resistant association between the translating ribosome and the thylakoid membrane is established (Zoschke and Barkan, 2015). This is most likely due to insertion of the nascent chain into the cpSec1/Alb3 membrane insertase machinery and contacts with the lipid bilayer. The transition from soluble to membrane-bound translating ribosomes needs to be orchestrated by many factors that are involved in nascent chain processing, insertion-competent folding, quality control, and sorting of the ribosome-nascent chain complexes (RNCs) to the target membrane. Chloroplasts harbor an extensive network of processing enzymes, molecular chaperones and proteases which together contribute to protein modification and maturation (Breiman et al, 2016; Nishimura et al, 2017; Ries et al, 2020; Sun et al, 2021; Trösch et al, 2015a; van Wijk, 2015). However, the question of which factors are specifically required for cotranslational thylakoid membrane protein biogenesis and how they exert their molecular function remains largely unexplored. Recently, the chloroplast trigger factor-like protein 1 (TIG1), a homolog of the bacterial chaperone trigger factor, that binds at the bacterial ribosomal polypeptide tunnel exit site and promotes folding of a broad subset of nascent chains, was shown to be partially associated with chloroplast ribosomes but its potential role as chaperone and its substrate specificity has not been defined yet (Olinares et al, 2010; Rohr et al, 2019). A recent proteome analysis of affinity-purified

[1]Molecular Biology of Plant Organelles, Faculty of Biology and Biotechnology, Ruhr University Bochum, Bochum, Germany. [2]Plant Biochemistry, Faculty of Biology and Biotechnology, Ruhr University Bochum, Bochum, Germany. [3]Max Planck Institute of Molecular Plant Physiology, Potsdam Science Park, Potsdam, Germany. [4]Physical Chemistry I, Faculty for Chemistry and Biochemistry, Ruhr University Bochum, Bochum, Germany. [5]Protein Crystallography, Faculty of Biology and Biotechnology, Ruhr University Bochum, Bochum, Germany. [6]Molecular Photosynthesis, Faculty of Biology, Heinrich Heine University Düsseldorf, Düsseldorf, Germany. [7]These authors contributed equally: Dominique S Stolle, Lena Osterhoff. ✉E-mail: danja.schuenemann@rub.de

chloroplast ribosomes from the green alga *Chlamydomonas reinhardtii*, demonstrated that Hsp70B, Hsp90C and the chaperonin Cpn60 are associated with translating ribosomes (Westrich et al, 2021). Factors involved in sorting of RNCs to the thylakoid membrane comprise the chloroplast 54 kDa signal recognition particle subunit (cpSRP54), which is homologous to cytosolic SRP54 targeting factors in eukaryotes and prokaryotes (Akopian et al, 2013; Ziehe et al, 2018). Substrate proteins that depend on cpSRP54 for efficient cotranslational targeting comprise central photosynthetic proteins, such as the core proteins of photosystem I (PSI) and photosystem II (PSII) (PsaA, PsaB, D1, D2) or the cytochrome $b_6f$ complex subunit PetB (Hristou et al, 2019). CpSRP54 can bind directly to the ribosome via the ribosomal subunit uL4c and supports targeting at early stages of translation (Hristou et al, 2019). CpSRP54 is also able to contact the nascent chain of D1 after its emergence from the ribosomal exit tunnel (Nilsson et al, 1999; Nilsson and van Wijk, 2002). An alternative cotranslational targeting mechanism was described for the chloroplast-encoded cytochrome $b_6f$ subunit cytochrome *f*. Cytochrome *f* is synthesized with a cleavable N-terminal signal peptide and is targeted by chloroplast SecA (cpSecA), a homolog of the bacterial ATPase SecA, which mediates targeting of preproteins to the plasma membrane Sec machinery for translocation (Fernandez, 2018; Röhl and van Wijk, 2001; Voelker and Barkan, 1995; Voelker et al, 1997; Zoschke and Barkan, 2015). Consistently, cpSecA was identified in the ribosome-associated proteome of *Chlamydomonas reinhardtii* chloroplasts (Westrich et al, 2021).

In addition to folding, quality control, and sorting, ribosome-associated factors are also important for the precise regulation of protein synthesis in response to changing environmental and internal cellular conditions. They include translation factors, ribosome recycling, hibernation and biogenesis factors as well as transcript-specific RNA binding proteins (Bohne et al, 2013; Kannangara et al, 1997; Sun and Zerges, 2015; Trösch and Willmund, 2019; Zoschke and Bock, 2018). Furthermore, there is growing evidence that metabolic enzymes can physically interact with components of the translation machinery. In a recent study, it was found that various enzymes, e.g., of carbon, amino acid, or chlorophyll metabolism, are associated with isolated chloroplast ribosomes (Westrich et al, 2021; Trösch et al, 2022). In line with this, previous studies have demonstrated a direct association of an enzyme of chlorophyll biosynthesis with chloroplast ribosomes and an enzyme of fatty acid biosynthesis with D1-encoding mRNA, affecting its localized translation (Bohne et al, 2013; Kannangara et al, 1997).

While previous studies made progress in deciphering the ribosome interactome, the specific proteome associated with a ribosome depending on the nascent chain and respective chloroplast ribosome heterogeneity remains unexplored.

Here, we present a method for the in vitro generation and affinity purification of stalled chloroplast ribosomes translating either different chain lengths of the thylakoid membrane protein D1 or the soluble ribosomal subunit uS2c as a control, using a chloroplast-derived homologous translation system. Mass spectrometry-based proteome analysis of the purified RNCs revealed a set of proteins specifically associated with D1-translating ribosomes including factors involved in nascent chain modifications and sorting as well as metabolic enzymes. Among these, we identified STIC2 as a novel ribosome-associated factor

and propose that STIC2 cooperates with cpSRP54 in cotranslational delivery of D1 and potentially other chloroplast-encoded photosynthetic subunits to the thylakoid membrane.

# Results

## Production and purification of stalled ribosome complexes translating D1 and uS2c nascent chains

The production and isolation of stalled chloroplast RNCs containing varying chains builds on a *Pisum sativum* (*P. sativum*) chloroplast-derived in vitro translation system using truncated mRNAs lacking a stop codon, which was previously used to generate stable RNC translation intermediates of D1 (Nilsson et al, 1999; Nilsson and van Wijk, 2002; Walter et al, 2015). Here, we adapted the method by incorporating a Twin-Strep-tag (TST) coding sequence (CDS) into the D1-encoding *psbA* mRNA allowing specific isolation of D1 RNCs by affinity purification. The TST was positioned between amino acids 25 and 26 of the D1 N-terminus (Fig. 1A) as the integrity of the *psbA* 5' untranslated region (UTR) and *psbA* coding region is crucial for translation initiation (Nakamura et al, 2016). To identify factors specifically associated with D1-translating ribosomes, ribosome complexes translating the soluble 30S ribosomal subunit uS2c were generated for comparison. Here, the TST was positioned between amino acids 17 and 18 of the uS2c peptide (Fig. 1B). To characterize possible differences in the ribosome interactome as a function of different translational stages, we used truncated *psbA* mRNAs of various lengths resulting in short, medium, or long nascent TST-D1 peptides, with a length of 56 and 69, 108 and 136, or 195 and 291 amino acid residues, respectively. The numbers correspond to the D1 amino acid sequence, and the peptides are hereafter referred to as TST-D1 (56) to TST-D1 (291). As the ribosomal exit tunnel accommodates ~40 residues, the short nascent chain constructs represent translational stages in which the first TMH of D1 is expected to be fully buried inside the tunnel, while the medium-length constructs are designed to expose the first TMH outside of the ribosomal tunnel. The long nascent chain constructs represent translational stages in which at least two TMHs are exposed from the ribosome (Fig. 1C). For uS2c, a translation intermediate encoding 158 amino acids of uS2c was generated, referred to as TST-uS2c (158) (Fig. 1D).

Previously, we demonstrated that D1 truncations generated by the homologous translation system are stably associated with ribosomes (Walter et al, 2015). To verify that the D1 truncations containing a TST and the newly generated uS2c truncation are also ribosome-associated, translation reactions of TST-D1 (136), TST-D1 (195), and TST-uS2c (158) were spun on 1 M sucrose cushions. Immunoblot analyses of the ultracentrifugation pellets using α-Strep antibodies and antibodies against the ribosomal subunit uL4 demonstrated that the nascent chains of D1 and uS2c cosedimented with the ribosomes through the sucrose cushion (Appendix Fig. S1A). Next, we examined, whether the Twin-Strep-tagged RNCs could be purified using Strep-Tactin-coupled magnetic beads. To this end, radiolabeled in vitro translation reactions of various TST-D1 truncations (56, 108, 136, 195, and 291) and TST-uS2c (158) were subjected to affinity purification. The untagged truncation D1 (195) was used as a control. As expected, the Twin-Strep-tagged D1 truncations and TST-uS2c (158) were detected in the eluates, while

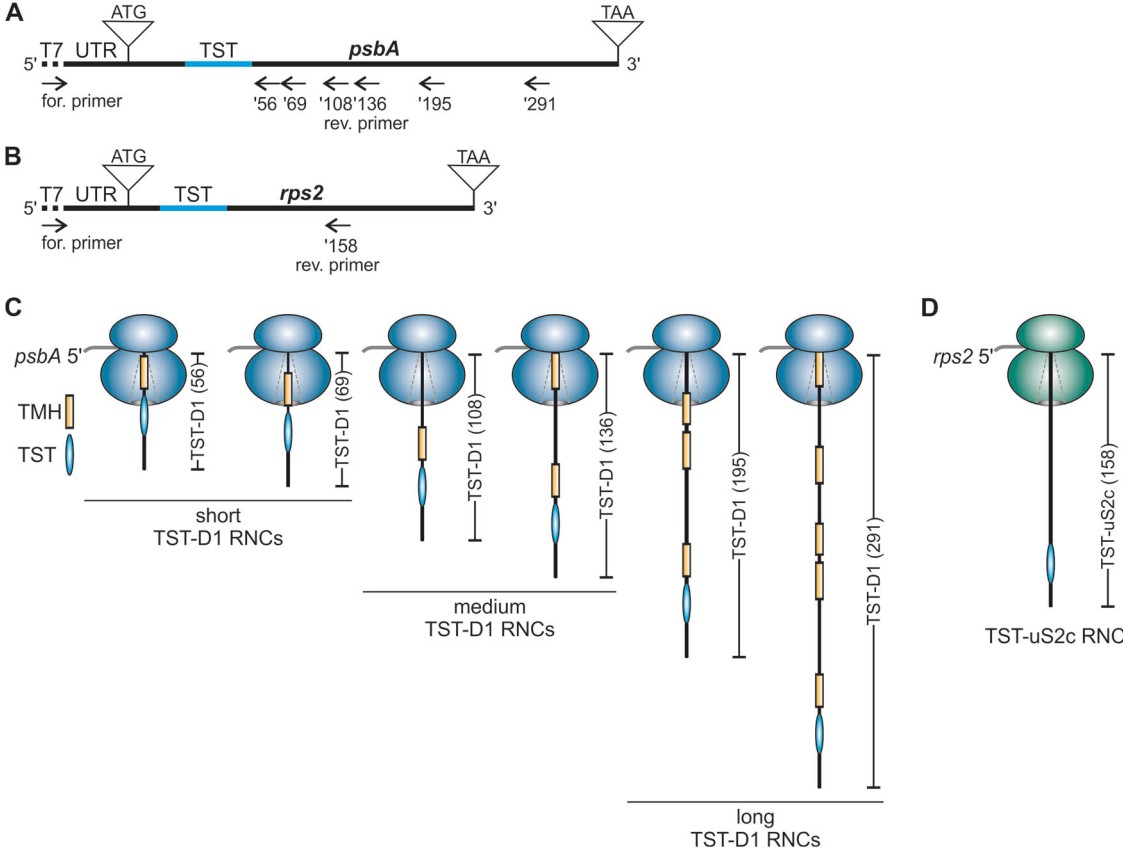

**Figure 1. Generation of stalled affinity-tagged ribosome-nascent chain complexes (RNCs).**

(**A**) Schematic representation of the *psbA* cDNA with T7 promotor sequence (dotted line) at the 5' end and the endogenous *psbA* 5' UTR (87 nt) used for PCR production of truncated templates for in vitro transcription into mRNA. A 90 nt Twin-Strep-tag (TST) coding sequence (blue line) was inserted at position +76 of the *psbA* sequence. The reverse primers ('56, '69, '108, '136, '195, and '291) lacking a stop codon, determine the length of the PCR product and are named according to the number of D1 amino acids encoded by the corresponding mRNA. (**B**) Scheme of the *rps2* cDNA used for PCR production of a truncated template for in vitro transcription into mRNA (description as in (**A**)). The endogenous 5' UTR comprises 90 nt. The TST coding sequence was inserted at position +52. The reverse primer ('158) lacks a stop codon and was used for PCR amplification of a product that corresponds to 158 amino acids of the uS2c protein. (**C, D**) Schematic RNCs generated by in vitro translation of mRNA from truncated templates as shown in (**A**) and (**B**). An internal TST is used for the affinity purification of the complexes. The nascent peptides of the D1 protein comprise at least one transmembrane helix (TMH) that is buried in the ribosome peptide tunnel or is exposed to the surrounding environment depending on the nascent peptide length. The nascent peptide of the soluble uS2c protein lacks any hydrophobic TMH.

the untagged D1 truncation remained in the supernatant (Appendix Fig. S1B).

## Mass spectrometric analysis of the purified RNCs

To identify factors associated with ribosomes translating D1, affinity-purified RNCs of the short, medium, and long TST-D1 intermediates and TST-uS2c (158) were applied to tandem mass spectrometry. Data were analyzed using MaxQuant and a *P. sativum* database (see Methods). *Arabidopsis thaliana* (*A. thaliana*) homologs were assigned using the best match in BLAST. A total of 259 proteins were identified (Dataset EV1). Of these proteins, 110 showed no significant quantitative differences in any of the short, medium, and long TST-D1 samples compared to the TST-uS2c (158) sample, whereas 141 proteins were significantly enriched (25 proteins) or exclusively detected (116 proteins) in at least one of the different chain length samples of D1 (Fig. 2A; Appendix Fig. S2, Dataset EV2, and Dataset EV3). Of the 141

putatively D1 RNCs-associated proteins, a core set of 46 proteins was enriched in all D1 chain length samples, while the remaining 95 proteins were assigned to either the short, medium, or long D1 chains or an overlap between two groups (Fig. 2B, Dataset EV3). As expected, peptides of D1 were exclusively detected in the D1 samples (Dataset EV3). As uS2c is a core ribosomal subunit, the detection of uS2c peptides in the uS2c sample as well as in D1 samples was also anticipated (Dataset EV2). However, peptides belonging to the Twin-Strep-tagged version of uS2c could only be detected in the uS2c sample. Chloroplast ribosomes consist of 57 core proteins, of which 24 compose the small 30S subunit and 33 the large 50S subunit (Ban et al, 2014; Bieri et al, 2016). Successful purification and comparative quantitative mass spectrometric analysis of the D1 and uS2c RNCs should therefore result in a large number of ribosomal proteins in the set of proteins showing no enrichment in the D1 versus the uS2c samples. In line with this, 26 core ribosomal proteins (8 subunits of the small subunit and 18 of the large subunit) were detected in the set of the 110

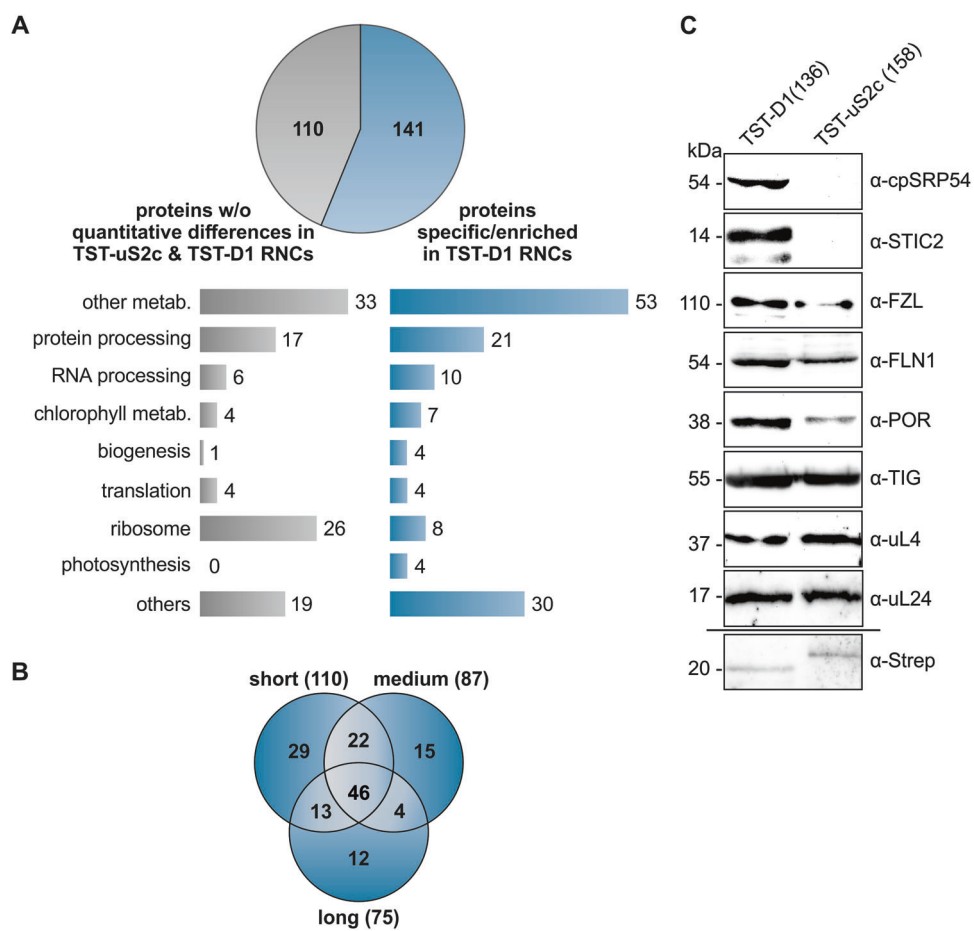

**Figure 2. Identification of proteins associated with affinity-purified stalled D1 RNCs.**

(A) Label-free quantitative tandem mass spectrometry identified 259 proteins in affinity-purified, stalled RNC complexes translating TST-uS2c (158) and TST-D1 of different lengths (TST-D1 (56), (69), (108), (136) (195), and (291)). Of these proteins, 141 were significantly enriched or exclusively present in TST-D1 RNCs, while 110 proteins showed no significant quantitative differences between the uS2c and D1 samples. Identified proteins were functionally categorized as indicated. The stalled RNCs were generated using the chloroplast-derived in vitro translation system and affinity-purified with MagStrep XT magnetic beads. (B) The set of 141 TST-D1 RNCs-associated proteins were visualized in a Venn diagram showing their distribution among the RNCs with short, medium, and long nascent TST-D1 peptides (as classified in Fig. 1) or overlaps of these groups. (C) Stalled RNCs of TST-D1 (136) and TST-uS2c (158) were generated using the chloroplast-derived in vitro translation system and purification was performed with StrepTrap XT columns. The purified complexes were subjected to immunoblot analysis using the indicated antibodies. The detection of the tagged D1 nascent chain using the α-Strep antibody was performed with the TST-D1 (195) variant because of a nonspecific signal of the α-Strep antibody at the size of TST-D1 (136). Source data are available online for this figure.

non-enriched proteins corresponding to 70% of all detected core ribosomal proteins (Fig. 2A; Appendix Fig. S2, Dataset EV4). Overall, 65% of the core ribosomal proteins were covered in the mass spectrometry dataset.

The 141 D1 RNCs-associated factors include 12 factors with uncharacterized function and 129 factors that can be classified into different categories covering protein/RNA processing, translation, biogenesis, and metabolisms (Fig. 2A).

Four factors were classified as biogenesis factors (Table 1). As expected, this group includes cpSRP54, of which ribosome association and function in cotranslational targeting has been demonstrated (Hristou et al, 2019; Nilsson et al, 1999; Nilsson and van Wijk, 2002; Piskozub et al, 2015; Walter et al, 2015). CpSRP54 was present in all nascent chain lengths of TST-D1 RNCs indicating that cpSRP54 associates with D1 translating ribosomes independent of the D1 chain length. Another biogenesis factor, that was specifically detected in the

short, medium, and long TST-D1 RNCs is the stromal suppressor of *tic40* protein 2 (STIC2) (Table 1). Recent data indicated that STIC2 and the thylakoid membrane protein Alb4, a homolog of the Alb3 insertase, cooperate in a cpSRP-dependent pathway in thylakoid membrane biogenesis (Bédard et al, 2017). Furthermore, we detected the fuzzy onions-like (FZL) protein, and the fructokinase-like 1 (FLN1) protein in two (small/medium) and all chain length samples, respectively (Table 1). Both proteins have been previously characterized as thylakoid membrane or chloroplast biogenesis factors (Borner et al, 2015; Gao et al, 2006; Gilkerson et al, 2012; Liang et al, 2018).

To verify that cpSRP54, STIC2, FZL, and FLN1 are specifically associated with D1 translating ribosomes, stalled translation products of TST-D1 (136) and TST-uS2c (158) were affinity purified and subjected to immunoblot analysis. In both samples, the equal abundance of ribosomes was demonstrated using antibodies against the ribosomal subunits uL4 and uL24 (Fig. 2C). The

**Table 1. Chloroplast proteins associated with D1 RNCs and functional in chloroplast or thylakoid biogenesis.**

| Uniprot ID | Uniprot names | Function | Psat | Potential AT | D1 RNC |
|---|---|---|---|---|---|
| P37107 | Signal recognition particle 54 kDa protein (cpSRP54) | Protein targeting | Psat3g032320.1 | AT5G03940.1 | s, m, l |
| O82230 | Suppressor of tic40 protein 2 (STIC2) | Thylakoid biogenesis | Psat1g022480.1 | AT2G24020.1 | s, m, l |
| Q1KPV0 | Probable transmembrane GTPase FZO-like (FZL) | Thylakoid biogenesis | Psat6g060000.3 | AT1G03160.1 | s, m |
| Q9M394 | Fructokinase-like 1 (FLN1; SCKL1) | Chloroplast biogenesis | Psat7g243520.1 | AT3G54090.1 | s, m, l |

Mass spectrometry analyses of purified TST-D1 RNCs vs. TST-uS2c RNCs showed a D1 specific enrichment of chloroplast biogenesis factors. All listed proteins were exclusively identified in TST-D1 RNCs and are predicted to contain a chloroplast targeting sequence according to Target-P 2.0. Peptide chain length of TST-D1 RNCs: short (s), medium (m), long (l). Detailed MS/MS analysis data are given in the Dataset EV1. Psat annotation was done according to *P. sativum* database as described in Methods.

**Table 2. Chloroplast proteins associated with D1 RNCs and functional in metabolic pathways.**

| Uniprot ID | Uniprot names | Function | Psat | Potential AT | D1 RNC |
|---|---|---|---|---|---|
| Q9FLW9 | Plastidial pyruvate kinase 2 (PKP2) | Carbohydrate | Psat1g096960.1 Psat6g194760.1 | AT5G52920.1 | $s^{(2.64)}$, $m^{(2.31)}$ s, m, l |
| Q43117 | Pyruvate kinase isozyme A (KPYA) | Carbohydrate | Psat6g113960.1 | AT3G22960.1 | $s^{(2.80)}$, $m^{(2.72)}$, $l^{(2.05)}$ |
| Q9ZU52 | Fructose-bisphosphate aldolase 3 (ALFP3) | Carbohydrate | Psat6g223200.1 | AT2G01140.1 | s, m, l |
| Q9SN86 | Malate dehydrogenase (MDHP) | Carbohydrate | Psat4g051000.1 | AT3G47520.1 | s, m, l |
| P12859 | Glyceraldehyde-3-phosphate dehydrogenase B (G3PB) | Carbohydrate | Psat5g010520.1 | AT1G42970.1 | s, m, l |
| Q9SZX3 | Argininosuccinate synthase (ASSY) | Amino acid | Psat0s3176g0120.1 Psat4g065240.1 | AT4G24830.1 | s, m, l $m^{(2.67)}$ |
| Q8L7R2 | Homoserine kinase (KHSE) | Amino acid | Psat2g000800.1 | AT2G17265.1 | s, m, l |
| Q93ZN9 | LL-diaminopimelate aminotransferase (DAPAT) | Amino acid | Psat7g104520.1 | AT4G33680.1 | s, m, l |
| Q94JQ3 | Serine hydroxymethyltransferase 3 (GLYP3) | Amino acid | Psat1g192400.2 | AT4G32520.1 | s, m, l |
| Q94AR8 | 3-isopropylmalate dehydratase large subunit (LEUC) | Amino acid | Psat6g107560.1 | AT4G13430.1 | s, m, l |
| Q1H537 | Divinyl chlorophyllide a 8-vinyl-reductase (DCVR) | Chlorophyll | Psat1g187280.1 | AT5G18660.1 | s, m, l |
| P45621 | Glutamate-1-semialdehyde 2,1-aminomutase (GSA) | Chlorophyll | Psat5g001920.1 | AT3G48730.1 | s, m, l |
| Q43082 | Porphobilinogen deaminase (HEM3) | Chlorophyll | Psat7g172760.1 | AT5G08280.1 | s, m, l |
| Q94A41 | Alpha-amylase 3 (AMY3) | Carbohydrate | Psat4g132240.1 | AT1G69830.1 | s, m, l |
| P52417 | Glucose-1-phosphate adenylyltransferase small subunit 2 (GLGS2) | Carbohydrate | Psat5g110720.1 | AT5G48300.1 | s, m, l |
| Q8W250 | 1-deoxy-D-xylulose 5-phosphate reductoisomerase (DXR) | Terpenoid | Psat4g058000.1 | AT5G62790.1 | s, m, l |
| Q94B35 | 4-hydroxy-3-methylbut-2-enyl diphosphate reductase (ISPH) | Terpenoid | Psat7g185320.1 | AT4G34350.1 | s, m, l |
| F6H7K5 | Thiamine thiazole synthase 2 (THI42) | Thiamine | Psat6g209120.1 | AT5G54770.1 | s, m, l |
| Q93X62 | 3-oxoacyl-reductase 1 (FABG1) | Fatty acid | Psat5g094480.1 | AT1G24360.1 | s, m, l |
| P52418 | Amidophosphoribosyltransferase (PUR1) | Purine | Psat7g181880.1 | AT2G16570.1 | s, m, l |

Proteins with functions in metabolic pathways for which an increased or exclusive association with all TST-D1 RNCs variants was detected by mass spectrometry are listed. Significant enrichment of proteins is indicated by LFQ intensity-based log2 values (D1/uS2c) given in parentheses (two-sided T-test, $p < 0.05$). For more details, see Table 1.

TST-labeled D1 and uS2c nascent chains were immunologically detected using α-Strep antibodies. Here, the longer chain length variant TST-D1 (195) was used because the α-Strep antibody showed a nonspecific signal that overlapped with TST-D1 (136). Notably, cpSRP54 and STIC2 were exclusively detected and FZL and FLN1 were clearly enriched in the TST-D1 (136) sample supporting the mass spectrometry data (Fig. 2C).

Recent reports indicated that chloroplast ribosomes exhibit a large interaction network that connects the protein synthesis machinery with various metabolic pathways (Trösch et al, 2022; Westrich et al, 2021). Consistently, among the 141 factors enriched in D1 RNCs, we identified 60 proteins that could be assigned to different metabolic pathways (Dataset EV3). As we considered an association of metabolic enzymes with ribosomes as a function of nascent chain length rather unlikely and to apply stringent parameters, we restricted this group to factors that were enriched in all chain length samples. This group comprises 21 proteins belonging to carbohydrate, amino acid, chlorophyll, terpenoid, thiamine, fatty acid, and purine metabolism (Table 2). Because of a potential link between chlorophyll metabolism and the biogenesis of chlorophyll-binding thylakoid membrane proteins such as D1 (Wang and Grimm, 2021) we focused on the chlorophyll biosynthesis enzymes. The mass spectrometry data indicated the specific presence of the glutamate-1-semialdehyde 2,1-aminomutase (GSA), the porphobilinogen deaminase (HEM3) and the divinyl chlorophyllide a 8-vinyl-reductase (DCVR) in all TST-D1 RNC samples (Table 2). Furthermore, the chlorophyll biosynthesis enzymes

**Table 3. Chloroplast proteins associated with D1 RNCs and functional in protein processing.**

| Uniprot ID | Uniprot names | Function | Psat | Potential AT | D1 RNC |
|---|---|---|---|---|---|
| Q9FUZ2 | Peptide deformylase 1B (PDF1B; DEF1B) | Deformylation, specificity to D1 | Psat3g166600.1 | AT5G14660.1 | s |
| Q9FV52 | Methionine aminopeptidase 1B (MAP1B) | N-terminal methionine excision | Psat2g085800.1 | AT1G13270.1 | $m^{(2.00)}$ |
| P08926 | 60 kDa chaperonin subunit alpha (CPN60A) | Chaperone | Psat7g144320.1 | AT2G28000.1 | $s^{(2.32)}$, $m^{(2.25)}$ |
| P08927 | 60 kDa chaperonin subunit beta (CPN60B) | Chaperone | Psat1g001680.1 Psat6g016920.1 | AT1G55490 | $s^{(1.60)}$, $m^{(1.62)}$ $s^{(2.37)}$, $m^{(2.37)}$ |
| Q9C667 | 60 kDa chaperonin subunit beta 4 (CPNB4) | Chaperone | Psat2g080360.1 | AT1G26230.1 | m |
| Q9LHA8 | Heat shock protein 70-4 (HSP70-4) | Chaperone | Psat3g182640.1 | AT3G12580.1 | s, m |
| Q9SIF2 | Heat shock protein 90-5 (HSP90-5) | Chaperone | Psat2g102720.1 | AT2G04030.1 | $s^{(4.53)}$ |
| Q9LF37 | Chaperone protein ClpB3 | Chaperone | Psat1g069480.1 | AT5G15450.1 | s, m |
| G7L2M8 | GrpE protein homolog | Chaperone | Psat3g085320.3 | AT5G17710.1 | $s^{(1.77)}$, $l^{(1.63)}$ |
| O22265 | Signal recognition particle 43 kDa protein (cpSRP43) | Chaperone | Psat7g178840.1 | AT2G47450.1 | s, m, l |
| P35100 | Chaperone protein ClpC (CLPC) | Chaperone | Psat5g054560.2 | AT5G50920.1 | $l^{(2.11)}$ |
| Q94B60 | Clp protease proteolytic subunit 4 (CLPP4) | Protease | Psat2g142240.1 | AT5G45390.1 | s, m, l |
| Q9S834 | Clp protease proteolytic subunit 5 (CLPP5) | Protease | Psat6g161200.1 | AT1G02560.1 | s, m, l |
| O82261 | Protease Do-like 2 (DEGP2) | Protease | Psat6g167960.1 | AT2G47940.1 | s, m |
| Q9LJL3 | Presequence protease 1 (PREP1) | Protease | Psat4g213560.1 | AT3G19170.1 | s, m |
| Q94AM1 | Organellar oligopeptidase A (OOPDA) | Protease | Psat2g133960.1 | AT5G65620.1 | s, l |
| Q8VZF3 | Probable glutamyl endopeptidase (CGEP) | Protease | Psat3g019800.3 | AT2G47390.1 | s |
| O64730 | Probable protein phosphatase 2C 26 (P2C26) | PSII phosphatase | Psat5g153360.1 | AT2G30170.1 | s, m, l |

Protein processing factors with increased or exclusive association with TST-D1 RNCs according to mass spectrometry analyses. Significant enrichment of proteins is indicated by LFQ intensity-based log2 values (D1/uS2c) given in parentheses (two-sided T-test, $p < 0.05$). For more details, see Table 1.

delta-aminolevulinic acid dehydratase (HEM2), coproporphyrinogen-III oxidase (HEM6), magnesium protoporphyrin IX methyltransferase (CHLM) and protochlorophyllide reductase (POR) were enriched in samples covering one or two chain lengths (Dataset EV3). The enrichment of POR in stalled D1 RNCs compared to uS2c RNCs was confirmed using immunoblot analysis (Fig. 2C).

A large functional D1 RNCs-associated group is formed by factors involved in protein processing like maturation, folding, or degradation (Table 3). Notably, among these are two factors of D1 nascent peptide maturation: the peptide deformylase PDF1B, which was exclusively found in short TST-D1 translation intermediates, and the subsequent acting methionine aminopeptidase MAP1B, which was enriched in D1-RNCs with medium long chains. The plastidic trigger factor (TIG1), a homolog of the bacterial trigger factor located close to the exit of the ribosomal polypeptide tunnel and acting as a molecular chaperone on emerging nascent chains, was detected in all TST-D1 RNCs as well as in TST-uS2c RNCs, but did not show an enrichment in any sample (Dataset EV1). Consistent with the mass spectrometry data, immunoblot analysis using purified TST-D1 (136) and TST-uS2c RNCs confirmed that both RNC types contained a similar amount of TIG1 (Fig. 2C). Therefore, our data suggest that the ribosome-associated protein biogenesis factors PDF1B, MAP1B, and cpSRP54 are specifically recruited to D1 translating ribosomes and that PDF1B and cpSRP54 can bind simultaneously to D1 translating ribosomes with nascent chains of up to 69 amino acids.

Moreover, members of the chloroplast chaperonin system, CPN60α and CPN60β subunits, were enriched in the short and medium D1-chain RNCs (Table 3). Chloroplast chaperonin CPN60 was originally described as a Rubisco-binding protein and a posttranslationally acting chaperone (Zhao and Liu, 2017).

However, it has also been described that CPN60 and its bacterial counterpart, the GroEL/GroES system, bind ribosomes in a puromycin-dependent manner pointing towards a cotranslational function (Kramer et al, 2019; Ries et al, 2020; Westrich et al, 2021). Additional chaperones that were enriched in one or more chain length samples are members of the HSP100, HSP90, and HSP70 molecular chaperone systems (HSP70-4, HSP90-5, CLPB3, CLPC, GrpE) and the 43 kDa subunit of the chloroplast SRP (cpSRP43) (Table 3). As factors that might be involved in degradation of aberrantly folded nascent D1-chains we detected various proteases (CLPP4, CLPP5, DEG2, PREP1, OOPDA, and CGEP) (Table 3).

## STIC2 and cpSRP54 are cooperatively required for the accumulation and maintenance of photosystem II

As described above, a comparative mass spectrometric analysis of purified RNCs translating either the membrane-integral D1 or the soluble uS2c protein identified STIC2 as a specific interactor of D1 RNCs. This finding, along with the known genetic interaction between a *stic2* mutation (*stic2-3*) and a *cpsrp54* mutation (*cpsrp54-3*), as well as STIC2's ability to physically interact with Alb3 and Alb4 (Bédard et al, 2017) led us to hypothesize that STIC2 plays a role in the cotranslational targeting and/or insertion of D1.

To test this hypothesis, we determined the maximum quantum yield (Fv/Fm) of PSII in 9-week-old *A. thaliana* plants grown under 50 µmol photons m$^{-2}$ s$^{-1}$ in a 12 h photoperiod. Wild type (WT) and mutants lacking either STIC2 (*stic2-3*) or cpSRP54 (*ffc1-2*) (Amin et al, 1999; Bédard et al, 2017), as well as a *ffc1-2 stic2-3* double mutant generated by crossing the single mutants, were measured (Fig. 3A; Appendix Fig. S3A,B). As previously described, the *stic2-3* mutant

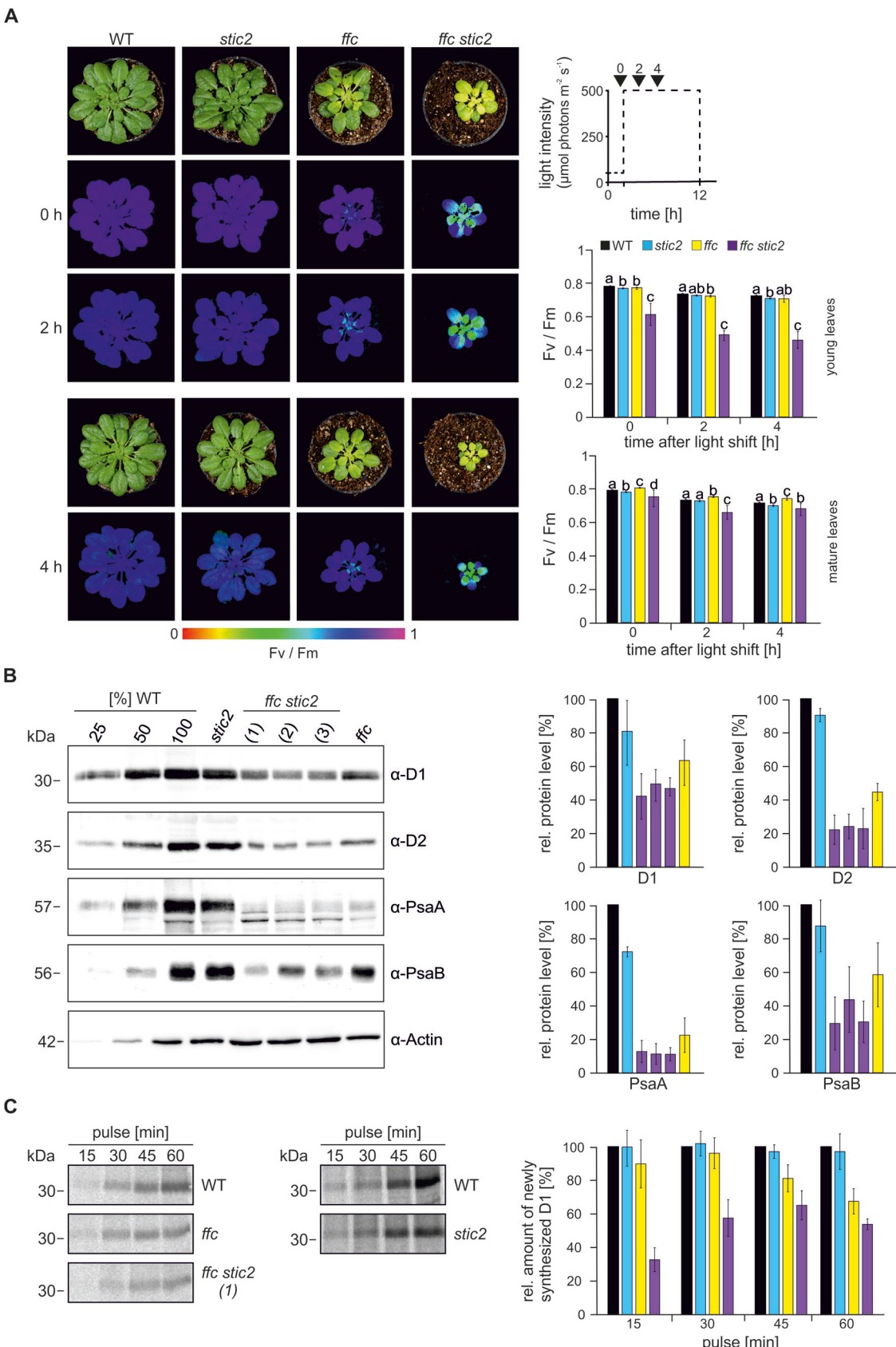

**Figure 3.  The combined loss of STIC2 and cpSRP54 causes high light sensitivity and low accumulation of D1.**

(A) Left: 9-week-old plants of wild type (WT), *stic2-3* (*stic2*), *ffc1-2* (*ffc*) and *ffc1-2 stic2-3* (*ffc stic2*) grown at 50 µmol photons m$^{-2}$ s$^{-1}$ with a 12 h photoperiod were shifted to high light (500 µmol photons m$^{-2}$ s$^{-1}$). The maximum quantum yield of PSII (Fv/Fm) was determined before the shift (0 h) and 2 and 4 h after the shift. Signal intensities for Fv/Fm are indicated by the false color scale at the bottom of the figure. Right: Overview of experimental setup indicating the time points for Fv/Fm measurements (0, 2, and 4 h) relative to the photoperiod and the light intensity (dotted line) (top). The maximum quantum yield of PSII (Fv/Fm) from young (middle) and mature (bottom) leaves of plants shown in the left panel. Averages and standard deviation are shown ($n = 4$–10). Statistically significant differences of the means ($p < 0.05$) between genotypes for the individual time point indicated by letters were determined by the Games-Howell multiple comparison test. (B) Total protein extracts from 5-week-old wild type (WT), *stic2-3*, *ffc1-2* and three lines of *ffc1-2 stic2-3* ((1), (2), (3)) were separated by SDS-PAGE and blotted for immunodetection with the indicated antibodies. The Actin level was monitored as loading control. The WT samples corresponded to 25, 50, and 100% of total protein. The protein levels were quantified using ImageJ in relation to WT (100%). Means and SDs were calculated from at least three independent biological replicates. (C) Leaf discs of wild type (WT) and the indicated *A. thaliana* mutants were incubated in a [$^{35}$S]-methionine containing solution in presence of cycloheximide at ambient light. After an incubation for 15, 30, 45, and 60 min thylakoid membrane proteins were extracted and used for SDS-PAGE and phosphor imaging. Signals were quantified (ImageJ) in relation to WT with WT corresponding to 100% for each time point. Means and SDs were calculated from at least three independent biological replicates. Source data are available online for this figure.

exhibits no visible phenotype, whereas the *ffc1-2* mutant shows reduced growth and a chlorotic phenotype with leaves becoming greener during development, albeit not to the same degree as WT (Amin et al, 1999; Bédard et al, 2017; Yu et al, 2012) (Fig. 3A). The *ffc1-2 stic2-3* double mutant is much smaller and more chlorotic than the single mutants, with the chlorotic phenotype most pronounced in young developing leaves (Fig. 3A). This is consistent with the *stic2-3 cpsrp54-3* mutant line described in Bédard et al (2017). The young *ffc1-2 stic2-3* leaves showed a clear decrease in Fv/Fm (0.61 in the mutant vs. 0.78 in WT), whereas mature leaves were only slightly impaired (0.75 in the mutant vs. 0.78 in WT, Fig. 3A). In contrast, the young and mature leaves of the single mutants exhibited Fv/Fm values comparable to WT (*ffc1-2* 0.77 and 0.80, *stic2* 0.77 and 0.78 for young and mature leaves respectively). High light causes photo-oxidative damage of the PSII D1 protein, which is accompanied by increased de novo synthesis of this subunit during a repair cycle (Jarvi et al, 2015; Nickelsen and Rengstl, 2013). If STIC2 functions together with cpSRP54 in the insertion of de novo synthesized D1 into the thylakoid membrane, we would expect high light treatment to exacerbate the Fv/Fm phenotype particularly of the double mutant. Therefore, we analyzed the sensitivity of the single and double mutants to high light. The application of high light (500 µmol photons m$^{-2}$ s$^{-1}$) for two or four hours resulted in a pronounced additional reduction of Fv/Fm in the young leaves of the *ffc1-2 stic2-3* mutant (0.46 in the mutant vs. 0.72 in WT after 4 h), while the mature leaves were slightly impaired (0.66 in the mutant vs. 0.71 in WT). The response of the young and mature leaves of the single mutants to high light was comparable to WT, showing Fv/Fm values between 0.70 and 0.74 after 4 h of high light (Fig. 3A).

To assess the steady state level of D1 and other cotranslationally targeted reaction center subunits of PSII (D2) and PSI (PsaA and PsaB) in the mutants, total leaf protein extracts of 5-week-old plants grown at 120 µmol photons m$^{-2}$ s$^{-1}$ in an 8 h photoperiod were subjected to immunoblot analysis and compared to WT (Fig. 3B). Consistent with our earlier findings (Hristou et al, 2019), the *ffc1-2* plant showed a ~40–75% reduction in the levels of the reaction center proteins. Reduced levels of D1, D2, PsaA, and PsaB were also observed in the *stic2-3* single mutant, but this reduction was much less pronounced, amounting to only ~13–25%. Notably, the *ffc1-2 stic2-3* double mutant was severely affected and showed the lowest levels of the reaction center proteins among the tested plant lines (Fig. 3B).

To evaluate whether the severely reduced steady state level of D1 and the light sensitivity observed in the *ffc1-2 stic2-3* mutant are due to altered synthesis or degradation rates of newly formed D1, we

conducted pulse-labeling/chase experiments using $^{35}$S-methionine on leaf discs. Pulse-labeling for 15, 30, 45, and 60 min showed substantially decreased accumulation of membrane-associated D1 in *ffc1-2 stic2-3*. The *ffc1-2* mutant was less affected, while *stic2-3* exhibited a D1 synthesis levels comparable to WT (Fig. 3C). Pulse-chase experiments showed no pronounced differences in D1 degradation between the mutants and WT, indicating that the low accumulation of D1 in *ffc1-2 stic2-3* is due to impaired synthesis of D1 (Appendix Fig. S4A). To investigate whether the altered D1 synthesis affects the assembly of PSII, the migration behavior of D1 and CP43 in solubilized thylakoids was analyzed by 2-D Blue-Native/SDS-PAGE and immunoblots. However, no differences in assembly were detected between WT, the *ffc1-2 stic2-3* double mutant, and the single mutants (Appendix Fig. S4B). In all genotypes, D1 and CP43 could be assigned to PSII assembly intermediates with the same relative distribution (PSII supercomplexes, dimeric PSII, monomeric PSII, CP43-free monomeric PSII).

Taken together, our data strongly suggest that STIC2 and cpSRP54 functionally cooperate in the de novo biogenesis of PSII during thylakoid membrane development and are also required for efficient D1 replacement in the PSII repair cycle under high light stress. Furthermore, our findings imply that the function of STIC2/cpSRP54 may not be restricted to D1, but may also affect the targeting of additional photosynthetic core proteins.

## STIC2 partially cofractionates with ribosomes and thylakoid membranes and affects translation in *Arabidopsis*

To validate the association of STIC2 with chloroplast ribosomes, cell extracts were prepared from freshly frozen leaf material of 5-week-old WT and *stic2-3* plants in presence of chloramphenicol to immobilize ribosomes on mRNA. The extracts were subjected to sucrose density gradient centrifugation, and the resulting fractions were analyzed by immunoblotting using antibodies against STIC2, cpSRP54 and the ribosomal subunits uL24 and uS5. Consistent with previous findings, cpSRP54 cosedimented with ribosomes at the bottom of the gradient, while a second pool remained in the upper fractions (Fig. 4A). This low molecular weight fraction of cpSRP54 presents the cpSRP43/cpSRP54 heterodimer, which functions in posttranslational LHCP transport (Ziehe et al, 2018). We also detected small amounts of the ribosomal proteins uL24 and uS5 in the upper fractions, indicating the presence of unassembled ribosomal subunits in this region of the gradient.

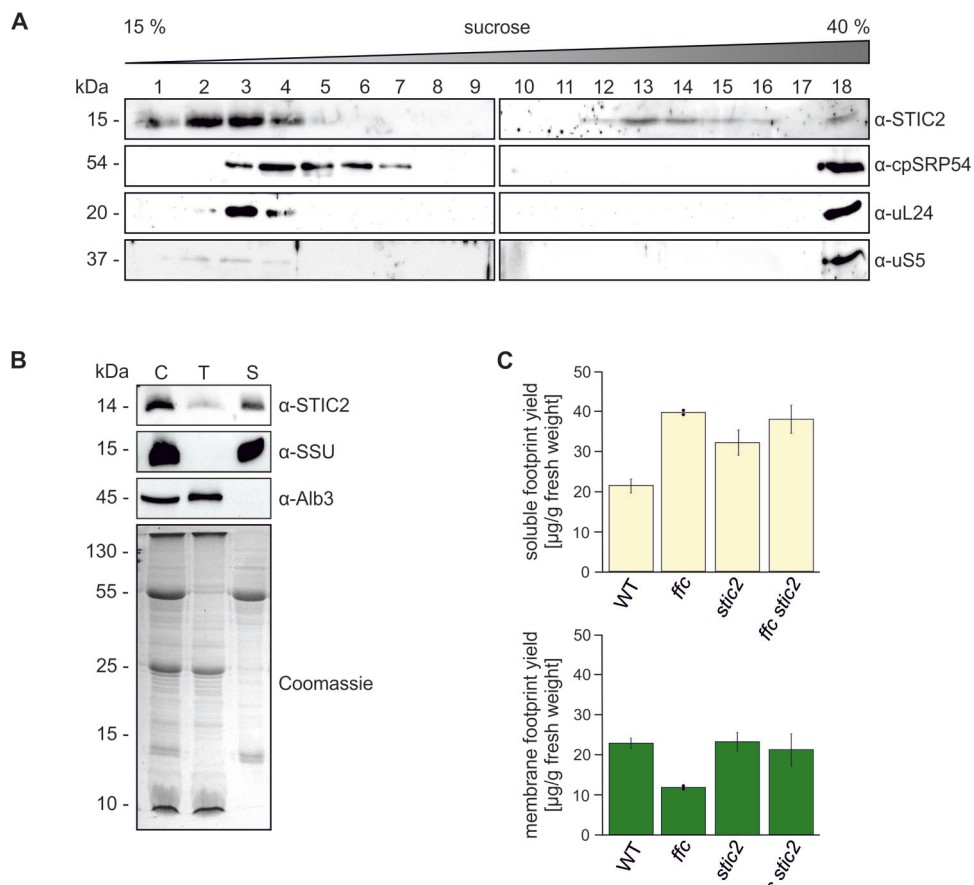

**Figure 4. Analysis of cofractionation of STIC2 with stromal ribosomes and thylakoid membranes and the effect of the *stic2* mutation on ribosomal footprints.**

(A) The interaction of endogenous STIC2 and ribosomes was tested by sucrose density gradient centrifugation of leaf extracts of *A. thaliana* WT plants. The gradient fractions were separated by SDS-PAGE and applied to immunoblotting using the indicated antibodies. A representative experiment from two independent biological and two technical replicates is shown. (B) Isolated chloroplasts from *A. thaliana* WT were lysed and separated into stromal and thylakoid fractions by centrifugation. Samples equivalent to 5 µg chlorophyll of chloroplast extract (C), washed thylakoids (T), and stroma (S) were separated on SDS-PAGE and blotted for immunodetection using antisera raised against STIC2, Alb3 and the small subunit of Rubisco (SSU). The Alb3 insertase and SSU were used as controls for successful fractionation. A Coomassie blue stained SDS gel served as loading control. A representative experiment from two independent biological replicates is shown. (C) Ribosome footprint yield was determined for soluble (yellow) and membrane (green) fractions of WT, *stic2-3*, *ffc1-2*, and *ffc1-2 stic2-3* and normalized to the amount of fresh weight used as starting material. Means and SDs are derived from two (*ffc1-2*) or three (WT, *stic2-3*, *ffc1-2 stic2-3*) independent biological replicates. Source data are available online for this figure.

The top fractions contained also most of the endogenous STIC2. Notably, a small but clearly detectable amount of STIC2 cofractionated with cpSRP54 and ribosomes at the bottom of the gradient (Fig. 4A). When using an extract from the *stic2-3* mutant, the ribosomal proteins and cpSRP54 exhibited the same running behavior as in WT, while no signal was obtained for STIC2, demonstrating the specificity of the STIC2 antibody (Appendix Fig. S4C). These data corroborate that STIC2 is partially associated with chloroplast ribosomes.

STIC2 was reported to be exclusively located in the chloroplast stroma (Bédard et al, 2017). However, earlier chloroplast sub-proteome studies (Peltier et al, 2004), along with evidence of STIC2 interacting with Alb3 and Alb4 (Bédard et al, 2017) and our finding that STIC2 is associated with D1 RNCs, suggest a partial or transient association of STIC2 with the thylakoids. To reexamine the subchloroplast localization of STIC2, *A. thaliana* chloroplasts were fractionated into stroma and thylakoids. In the subsequent immunoblot, STIC2 was detected predominantly in the stromal

fraction. However, a significant amount of the protein was also present in the thylakoid fraction (Fig. 4B) supporting a function of STIC2 in cotranslational insertion of D1 and potentially other photosynthetic proteins into the thylakoid membrane.

To further investigate the role of STIC2 in translation and cotranslational sorting, we quantitatively assessed the yield of membrane-associated and soluble ribosomal footprints in *stic2-3* and the *ffc1-2 stic2-3* double mutant, compared to *ffc1-2* and WT. The experimental setup included a nuclease pretreatment of the membrane fraction to ensure that only ribosomes stably bound to the thylakoid membrane via direct contact with the insertase machinery were classified as membrane-associated (Zoschke and Barkan, 2015; Hristou et al, 2019). As previously shown (Hristou et al, 2019), the *ffc1-2* mutant exhibited a significant increase in soluble ribosomal footprints and a corresponding decrease in membrane-associated footprints, consistent with cpSRP54's role in forming a stable contact between the RNC and the membrane (Fig. 4C). Notably, the *stic2-3* mutant also displayed an altered ratio

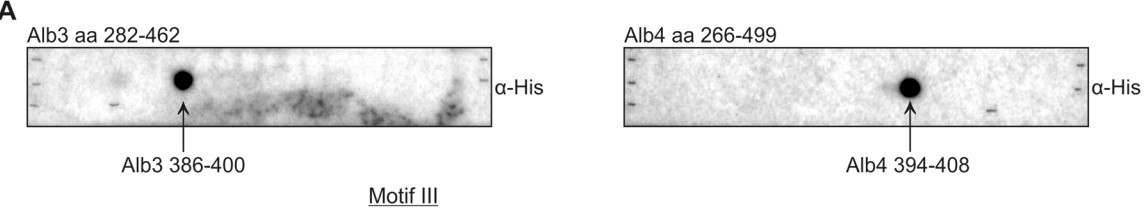

Motif III

```
Alb3 375  AKRSIAQPDDA**GERFRQLKEQEKRSK**KNKAVA  406
Alb4 386  GEKVTPEC**PKPGERFRLLKEQEA**KRRREKEER  417
```

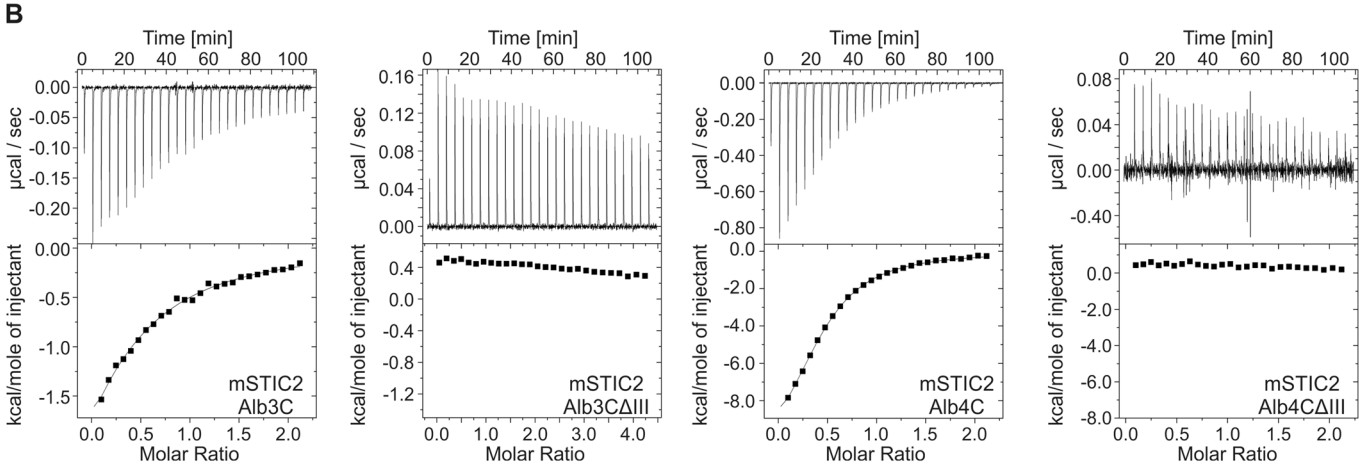

**Figure 5.    The conserved C-terminal Motif III of Alb3 and Alb4 is the primary interaction site for STIC2.**

(A) The interaction of His-STIC2 with the C-terminal regions of Alb3 (aa 282–462, left side) and Alb4 (aa 266–499, right side) was analyzed using pepspot-labeled nitrocellulose membranes. Recombinant His-STIC2 was incubated in a final concentration of 5 µg/ml with the pepspot membranes. Bound His-STIC2 was detected with antisera directed against the His-tag. Detected spots correspond to amino acids 386–400 of Alb3 and amino acids 394–408 of Alb4. These residues are indicated in bold in an alignment of a C-terminal region of Alb3 and Alb4 comprising the conserved Motif III (underlined). (B) Representative measurements of isothermal titration calorimetry (ITC) to determine binding affinities for the interaction of STIC2 with the C-terminal regions of Alb3 and Alb4. Left panels: 1.5 mM His-STIC2 was titrated into 0.15 mM of Alb3C-His or Alb3CΔIII-His. Right panels: 1 mM His-STIC2 was titrated into 0.1 mM Alb4C-His or Alb4CΔIII-His solutions. The resulting changes in heating power were recorded (top panels). After integration, the resulting enthalpy changes are plotted versus the molar ratio of STIC2 and the indicated binding partners (bottom panels). The titration isotherms resulted in a Kd of 150 ± 30 µM for the STIC2/Alb3C interaction and a Kd of 17 ± 4 µM for the STIC2/Alb4C interaction. No binding was detected for STIC2/Alb3CΔIII and STIC2/Alb4CΔIII. Data were calculated from two to five independent experiments. Source data are available online for this figure.

of soluble-to-membrane footprint yield, characterized by a substantial increase in soluble footprints while maintaining a similar yield of membrane-associated footprints compared to WT (Fig. 4C). As expected, the *ffc1-2 stic2-3* double mutant showed a clear increase in soluble footprint yield. However, the yield of membrane-associated footprints was only slightly decreased compared to WT (Fig. 4C). While our footprint analysis allows for speculation regarding the precise molecular function of STIC2 (see discussion), the data reveal important aspects of STIC2's functional role. Our results show that STIC2 affects chloroplast translation and are, therefore, consistent with a role of STIC2 in the translation of cotranslationally sorted thylakoid membrane proteins. Furthermore, our data show that, unlike cpSRP54, STIC2 is not required for efficient tethering of the RNCs to the membrane.

## Molecular dissection of the interaction between STIC2 and the Alb3/4 proteins

Alb3 and Alb4 are involved in insertion, folding and assembly of thylakoid membrane proteins (Benz et al, 2009; Trösch et al, 2015b;

Wang and Dalbey, 2011). They are characterized by a conserved hydrophobic core region with five TMHs and a stromal-exposed positively charged extension at their C-termini (Ackermann et al, 2021; Hennon et al, 2015; McDowell et al, 2021). Previously, it has been suggested that a conserved sequence within the stromal C-terminal regions, known as Motif III (Falk et al, 2010), is important for STIC2 binding. This conclusion was based on the finding that a mutation of a conserved Gly (G397 in Alb4) within Motif III of the Alb4 C-terminus substantially reduced its binding to STIC2 (Bédard et al, 2017). To further analyze and systematically map the binding sites of full-length Alb3 and Alb4 for STIC2, we utilized pepspot analyses. For mature Alb3 (aa 56–462), the designed pepspots covered the first stromal loop (aa 155–208) and the region from TMH3 to the C-terminus (aa 282–462) in 15mer peptides with an overlap of 11 aa. Corresponding homologous structures of Alb4 were covered with similarly designed pepspots ranging from aa 139–192 and aa 266–499 (Fig. 5A; Appendix Fig. S5). Incubation with recombinant mature His-STIC2 and subsequent detection with His-tag specific antibodies revealed that STIC2 interacted with one peptide of Alb3 (aa 386–400) and one

peptide of Alb4 (aa 394–408) (Fig. 5A). These peptides largely correspond to the conserved Motif III in Alb3 and Alb4, corroborating the results of Bédard et al (2017). Furthermore, our data proof a direct binding between Motif III and STIC2 and strongly suggest that Motif III forms the primary contact site.

To quantitatively analyze the interactions between STIC2 and the C-terminal region of Alb3 and Alb4, the binding reactions were assessed using isothermal titration calorimetry (ITC). Titrations of His-STIC2 into Alb3C-His yielded an average dissociation constant of 150 μM (±30 μM) (Fig. 5B). In contrast, an average dissociation constant of 17 μM (±4 μM) was determined for His-STIC2 and Alb4C-His (Fig. 5B). The constructs Alb3CΔIII-His and Alb4CΔIII-His showed no binding to STIC2 (Fig. 5B). Our data indicate that Alb4 has approximately ninefold higher affinity for STIC2 compared to Alb3. This suggests that Alb4 is the primary interactor of STIC2, although further research on the physiological concentrations of Alb3 and Alb4 is necessary to support this conclusion.

To further characterize the STIC2 binding interface with Alb4C and Alb3C, we used AlphaFold-multimer as implemented in the Colabfold pipeline to model the structure of the complexes (Evans et al, 2021; Jumper et al, 2021; Mirdita et al, 2022). STIC2 was modeled as a homodimer with high confidence for the core region of the protein (Fig. 6A; Appendix Fig. S6A). One of the templates used by Colabfold is its bacterial homolog, YbaB, from *Haemophilus influenzae*, which forms a tightly associated dimer (Lim et al, 2003). STIC2 exhibits high structural similarity to YbaB with each monomer displaying a α1/β1/β2/β3/α'/α2 topology (Fig. 6A). For the C-termini of Alb3 and Alb4, no suitable structural templates were found by Colabfold. While the resulting models have no predictive value in the terminal regions, they have a high-quality score for one long α-helix, containing Motif III (Appendix Fig. S6A). Notably, this region is predicted to form the binding interface with STIC2, which is conserved in all five independent Alb4/STIC2 and Alb3/STIC2 models obtained by Colabfold (Fig. 6B,C; Appendix Fig. S6A). Since we observed no profound differences between the models for Alb3C or Alb4C, we subsequently focus on the Alb4C/STIC2 model.

Numerous residues in Motif III are in close proximity (≤4 Å) to STIC2 allowing for various contacts that enable the STIC2/Alb4C interaction (Fig. 6B). In the structural model, the main chain nitrogen atoms of G397 and R399 form hydrogen bonds to E112 of STIC2 leading to capping of the N-terminal end of the Motif III helix (Fig. 6C; Appendix Fig. S6B). The exact position of E112 seems to be dependent on the preceding flexible glycine G111, which is furthermore forming a polar contact to the Alb4C side chain of R399. The guanine group of R399 on Alb4C itself forms contacts to the backbone of STIC2 (Appendix Fig. S6B). Furthermore, F400 of Alb4C stacks to a hydrophobic patch on the surface of STIC2, which is formed by the apolar parts of the side chain of K115 of STIC2 (Fig. 6B; Appendix Fig. S6C). Another specific interaction is formed between E407 of Alb4C and the STIC2 backbone. In addition, Alb4C's R410 and R411 are predicted to form salt bridges to E131 of STIC2, which is positioned between the end of the third β-sheet and the beginning of the short α'-helix (Fig. 6C; Appendix Fig. S6D).

To validate the predicted interaction interface, we conducted pulldown experiments using Alb4C-His variants bearing the single point mutations R399G, F400G, E407G, and R410G. These variants

exhibited about 70–80% reduced binding to GST-STIC2 confirming that these Alb4C residues play a central role for STIC2 binding (Fig. 6D).

To analyze the roles of G111, E112, K115, and E131 in STIC2 for Motif III binding, GST-STIC2 point mutation variants were used in pulldown assays with Alb4C and Alb3C (Fig. 6E,F). The G111L mutation led to a reduction of about 50 to 60% in coelution with Alb4C and Alb3C (Fig. 6E,F). The STIC2 E112A mutation resulted in an about 65% decrease in binding Alb4C, while the K115A variant did not affect the interaction with Alb4C (Fig. 6E). Different to Alb4C, the E112 mutation in STIC2 only slightly impaired the interaction with Alb3C, reducing binding by about 15%, while the K115A variant led to an about 60% decrease in Alb3C binding. To further analyze the contribution of K115 and E112 in Alb4C and Alb3C binding, the STIC2 E112A K115A double mutant was generated. As shown in Fig. 6E and F, the double mutation resulted in a substantially reduced binding by about 65% and 80% to Alb4C and Alb3C, respectively. Lastly, the influence of E131 on the interaction with the C-termini of Alb4 and Alb3 was analyzed, revealing a marked binding reduction of about 80% to both proteins (Fig. 6E,F). In conclusion, our experimental approach mapped the interaction interface between STIC2 and the C-terminal regions of Alb4 and Alb3 to the negatively charged β-sheet region of STIC2 and Motif III and identified several residues in both interaction partners critical for binding.

## Discussion

In this work, we affinity-purified stalled in vitro translating chloroplast ribosomes via Twin-Strep-tagged nascent peptides of the PS II reaction center protein D1 and ribosomal uS2c. This method allows the purification of a homogeneous chloroplast ribosome subpopulation and thus the analysis of the interaction network of ribosomes specifically translating a thylakoid membrane protein. Mass spectrometric and immunoblot analyses of the purified RNCs showed that proteins from metabolic, biogenesis, and protein processing pathways associate with D1-translating ribosomes in the chloroplast (summarized in Fig. 7).

D1 binds chlorophyll a together with D2 as part of the reaction center of PSII required for the light induced charge separations. Based on the hazardous high chemical reactivity of photoexcited chlorophyll, it is expected that the chlorophyll demand during photosystem biogenesis at the thylakoid membrane is spatiotemporally coupled to its supply either by chlorophyll biosynthesis or from an existing pool of protein-bound chlorophyll. Indeed, in recent years, a chlorophyll assembly complex has been identified in cyanobacteria and plants, that is proposed to deliver newly synthesized chlorophyll to the nascent D1 chain during cotranslational insertion into the thylakoid membrane (Chidgey et al, 2014; Hey and Grimm, 2020; Knoppova et al, 2014; Li et al, 2019; Wang and Grimm, 2021) and to regulate the translation of D1 in land plants (Chotewutmontri and Barkan, 2020). In this work, we observed that several chlorophyll biosynthesis enzymes (GSA, HEM2, HEM3, HEM6, CHLM, POR, and DCVR) are specifically enriched in a ribosome population translating D1. Interestingly, a direct interaction between chloroplast ribosomes and a subunit of the magnesium-chelatase, another enzyme of the chlorophyll biosynthesis pathway, has been suggested by Kannangara et al

**A**

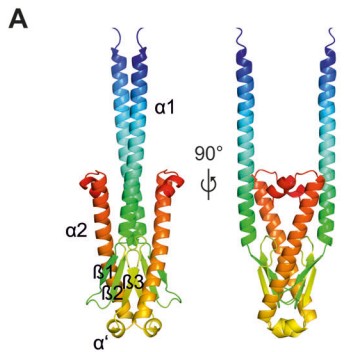

**B**

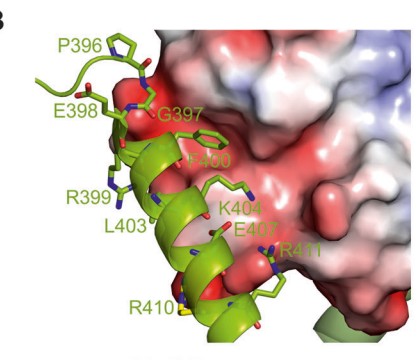

Motif III
Alb4 397 GERFRLLKEQEAKRRREK 414

**C**

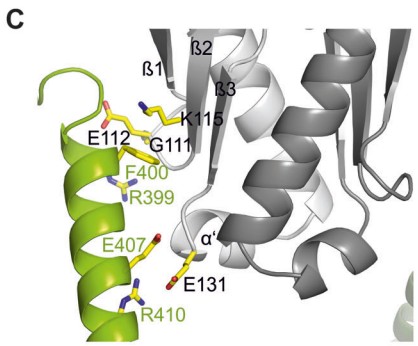

**D**

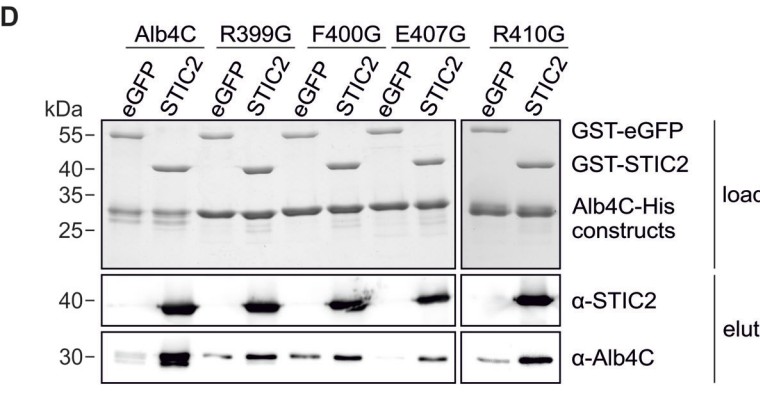

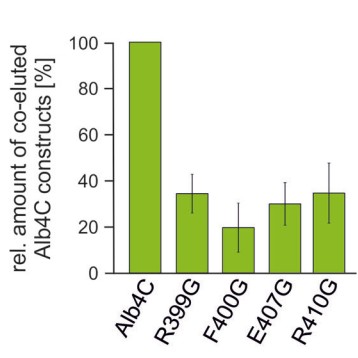

**E**

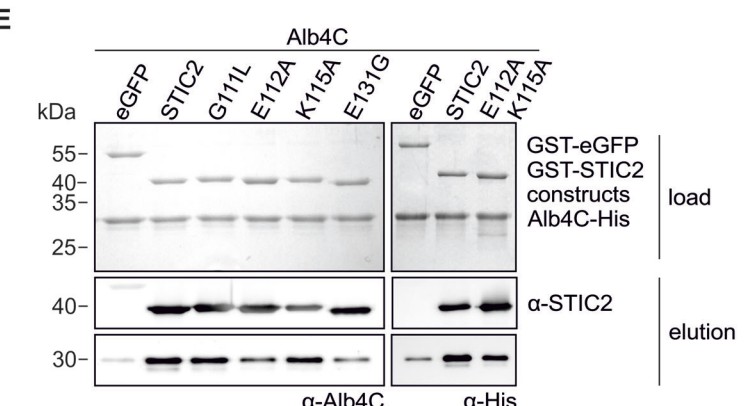

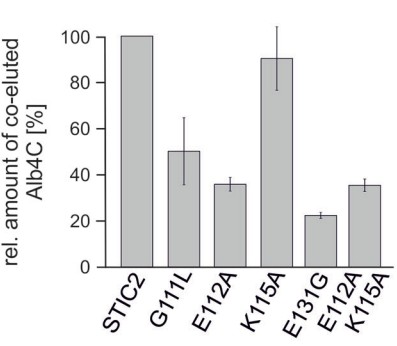

**F**

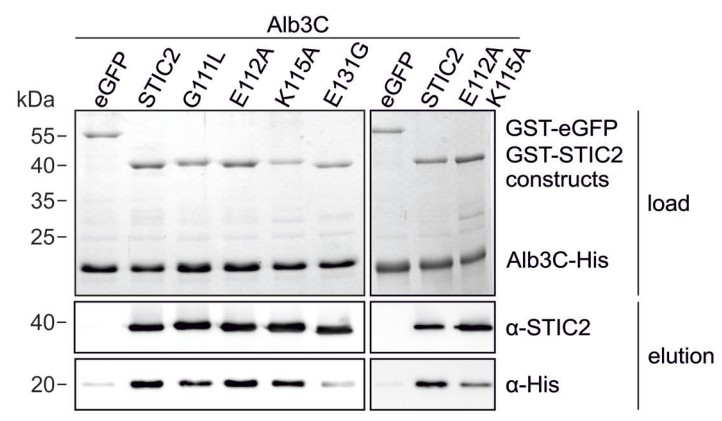

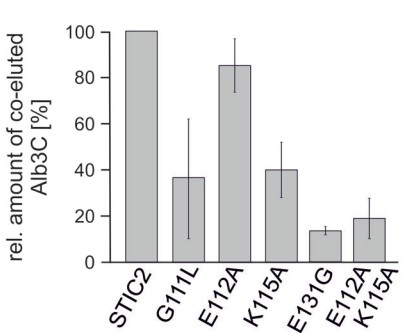

**Figure 6.   Characterization of the interaction interface between STIC2 and the C-terminal region of Alb4 and Alb3.**

(A) Alphafold-Multimer Model of the STIC2 dimer. STIC2 is shown in two orientations in cartoon representation with rainbow coloring from blue to red. (B) Interaction of Alb4 Motif III with STIC2. Shown is the STIC2 electrostatic surface (Connolly surface colored with APBS potential from red to blue (−4 to +4 kJ/mol/e)) together with a cartoon representation of Alb4C in green. Residues of Alb4 within 4 Å of STIC2 are shown as sticks and labeled. (C) Interacting residues investigated in the study. Shown is STIC2 as gray cartoon model (monomer A, light gray, monomer B darker gray), Alb4C as green cartoon. Interacting residues characterized experimentally are shown as yellow sticks and labeled. (D–F) Pull-down experiments using purified GST-STIC2 and Alb4C variants are shown in (D) or the indicated GST-STIC2 point mutation variants and Alb4C (E) or Alb3C (F). GST-eGFP was used as a control. Assays were conducted using glutathione sepharose beads. Load samples are shown by Coomassie stained SDS-Gels. Eluates were analyzed by SDS-PAGE and subsequent immunoblotting using the indicated antibodies. Coelution was quantified using ImageJ. Means and SDs were calculated from three independent experiments. Source data are available online for this figure.

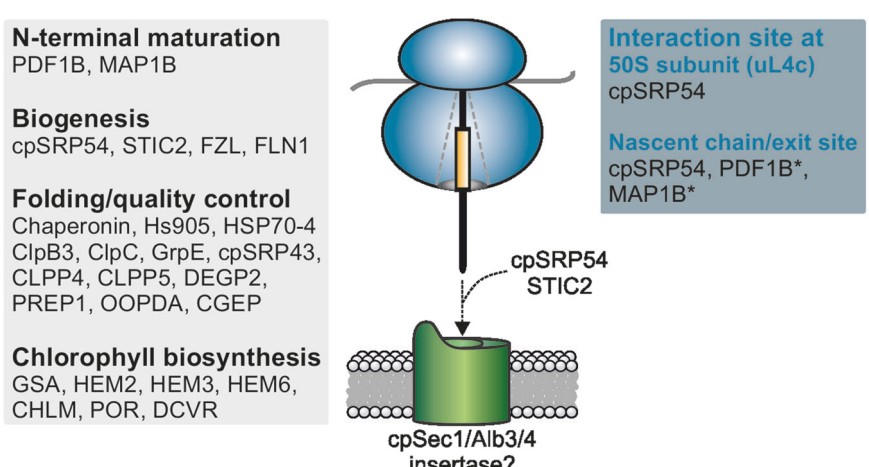

**Figure 7.   Chloroplast ribosome interactors during D1 synthesis.**

Summary of the interactors of D1 RNCs identified in this work, whose relation to D1 biosynthesis is discussed in the main text. The functional categories (gray box) and the interaction sites with RNCs (blue box) are indicated. The interaction of cpSRP54 with the ribosomal 50S subunit uL4c and nascent D1 peptides were described previously (Hristou et al, 2019, Nilsson et al, 1999, Nilsson and van Wijk, 2002). The ribosomal interaction sites of marked (*) factors of nascent chain maturation and chaperones are indicated according to the interaction sites for prokaryotic protein biosynthesis (see main text). The cooperative function of the two biogenesis factors, STIC2 and cpSRP54, is required for efficient D1 biogenesis. In this context, the interaction of STIC2 with the thylakoid membrane proteins, Alb3 and/or Alb4, might be important to ensure efficient coupling of translation and insertion of the nascent chain into the membrane by the insertase machinery, likely formed by a cpSec1/Alb3/4 complex.

(1997). Although the physiological role of an association of chlorophyll biosynthesis enzymes with ribosomes is currently unclear, it could be speculated that the translating ribosome might influence the activity of the biosynthetic enzymes or might coordinate the sites of chlorophyll biosynthesis and protein insertion at the thylakoid membrane.

Furthermore, we identified several metabolic enzymes of e.g., the carbon or amino acid metabolism in the D1 RNC interactome. As it has been shown that ribosomes play an important role in the control of metabolic acclimation (Garcia-Molina et al, 2020) our data suggest that the coordination occurs, at least in part, directly at RNC complexes. While our findings are consistent with a previous study identifying many metabolic enzymes in the ribosome interactome (Westrich et al, 2021), we should consider that the spatial clustering of sequentially acting metabolic enzymes might be affected in a translation system, potentially influencing local protein associations.

An interesting link between D1 synthesis and thylakoid membrane biogenesis arises from the association of the fuzzy onions-like (FZL) chloroplast membrane-remodeling GTPase to D1 RNCs. Notably, this link was previously suggested, as during early steps of thylakoid biogenesis accumulation of FZL at the thylakoids is accompanied by increased thylakoid binding of RNCs and insertion of plastid-encoded PS II subunits (Liang et al, 2018). FZL was also identified in the ribosomal interaction network in *Chlamydomonas reinhardtii* chloroplasts (Westrich et al, 2021) and was described in the regulation and determination of thylakoid and chloroplast morphology in plants (Gao et al, 2006).

Previous studies concluded that the transcriptional and translational system might be linked in chloroplasts (Pfalz et al, 2006; Zoschke and Bock, 2018). In support of these data, we observed an association of fructokinase-like 1 (SCKL1/FLN1) in all D1 chain length samples. As part of the plastid-encoded RNA polymerase, FLN1 participates in the transcription of photosynthesis-related genes (Borner et al, 2015; Shiina et al, 2005).

Protein translation in the cytosol as well as in organelles is linked to cotranslational events associated with processing and quality control of the nascent chains and mediated by a diverse protein network (Breiman et al, 2016; Ries et al, 2020; Trösch et al, 2015a; Westrich et al, 2021). In chloroplasts, N-terminal methionine modification is a widespread process because many proteins are synthesized with an N-terminal formylated methionine and undergo processing by either removal of the formylgroup or the formylmethionine (Giglione and Meinnel, 2001; Zybailov et al,

2008). A loss of N-terminal methionine deformylation by PDF1B leads to an increased degradation of photosynthetic proteins including D1 and D2, and an impaired chloroplast biogenesis and plant development (Adam et al, 2011; Giglione et al, 2003). In this work, we showed that PDF1B and the methionine aminopeptidase MAP1B are especially associated with D1 RNCs harboring nascent chains of less than 69 and 136 residues, respectively. These data strongly suggest that MAP1B works together with PDF1B in N-terminal methionine excision of D1. Moreover, our data indicate that at least deformylation occurs while the first TMH of D1 is fully buried in the ribosomal peptide tunnel and the N-terminus close to the peptide tunnel exit site, consistent with models of non-organellar N-terminal methionine excision processes (Breiman et al, 2016). In bacteria, multiple ribosome-associated protein biogenesis factors (e.g., PDF, MAP, SRP, and trigger factor) bind to ribosomal subunits near the exit site of the peptide tunnel with partially overlapping binding sites and their function on the nascent chain is finely coordinated in space and time (Kramer et al, 2019; Yang et al, 2019). Our results indicate that besides the methionine excision enzymes, the conserved protein biogenesis factors cpSRP54 and TIG1 are associated with D1 RNCs. However, the structural arrangement of biogenesis factors on the chloroplast ribosome is almost completely unknown. Recently, it was shown that cpSRP54 can contact the ribosome away from the peptide tunnel exit site by binding to the surface exposed domain of the ribosomal subunit uL4c (Hristou et al, 2019). However, the binding of cpSRP54 to ribosomes seems to be flexible as cpSRP54 is also able to contact the D1 nascent chain which indicates a position near the tunnel exit (Nilsson et al, 1999; Nilsson and van Wijk, 2002).

We also identified the 43 kDa subunit of the chloroplast SRP in the D1 RNC interactome. This was unexpected, because cpSRP43 functions in posttranslational transport of the light harvesting complex proteins and has not been identified as ribosome-associated before (Ziehe et al, 2018). However, cpSRP43 can displace cpSRP54 from the ribosome (Hristou et al, 2019) which might explain at least a transient ribosome association. Furthermore, recent studies showed that cpSRP43 acts as a chaperone for multiple enzymes of chlorophyll biosynthesis (Ji et al, 2021; Wang et al, 2018). As various chlorophyll metabolic enzymes were shown to be ribosome-associated (Trösch et al, 2022; Westrich et al, 2021; this work), it can be speculated that cpSRP43 might be indirectly bound to ribosomes via these enzymes.

Molecular chaperones as well as components of the chloroplast caseinolytic protease (Clp) system and other proteases were also enriched in D1 RNCs, suggesting their possible involvement in cotranslational protein biogenesis. However, we cannot rule out the possibility that incorporation of a Strep tag in the N-terminal region of D1 or translation in absence of the thylakoid may influence the folding or initiate degradation of the nascent chain. Therefore, the precise molecular and physiological functions of these factors in cotranslational folding or quality control remain unclear and need to be addressed in future studies.

In this study, we present compelling evidence that STIC2 cooperates with cpSRP54 in the cotranslational sorting of D1, an essential process for PSII biogenesis and repair. Several lines of evidence consistently support this functional role of STIC2. Mass spectrometric and immunological analysis of purified D1-translating ribosomes, generated by a chloroplast-based translation system, clearly demonstrated ribosome association of STIC2. This finding is further supported by the observation that a pool of STIC2 cofractionates with chloroplasts ribosomes in leaf extracts. Therefore, the intraorganellar localization of STIC2 is similar to cpSRP54, whose partial ribosome association has been demonstrated previously (Franklin and Hoffman, 1993; Schünemann et al, 1998; Hristou et al, 2019). Consistent with a role of STIC2 in translation, the stic2-3 mutant exhibits an increased yield of soluble ribosomal footprints. Although we cannot rule out that the loss of STIC2 results in an enhanced translation of chloroplast-encoded thylakoid membrane or stromal proteins, we consider it likely that the stic2 mutant is impaired in the elongation of nascent chains that are in close contact with the insertase machinery. Inhibition or slowdown of the cotranslational membrane insertion process would lead to an accumulation of ribosomes with nascent chains that are not sufficiently long for stable interaction with the thylakoid membrane. This model aligns with the observed increase in soluble footprints and the unaltered amount of membrane-bound footprints in the stic2-3 mutant. Unexpectedly, the double mutant shows only a slight decrease in membrane-bound footprints. This may imply that in absence of both STIC2 and cpSRP54, ribosome binding occurs through an alternative mechanism, that, however, does not support efficient cotranslational insertion. The interaction between ribosome-associated STIC2 and Alb4/Alb3 may be important for coupling of translation and insertion, ensuring the efficient biogenesis of thylakoid membrane proteins. The loss of STIC2 significantly exacerbates the D1 biogenesis/repair defect of a mutant lacking cpSRP54, further supporting the role of STIC2 in cotranslational sorting of D1. Based on the severe reduction of PsaA and PsaB in the ffc1-2 stic2-3 double mutant, it seems likely that STIC2 plays a general role in the cotranslational sorting of plastid-encoded multi-span proteins.

We observed that STIC2 binds with a ninefold higher affinity to Alb4C compared to Alb3C. Although the positively charged Motif III in both Alb4C and Alb3C is highly conserved (Falk et al, 2010), their amino acid sequence does not share full identity, which may contribute to the different binding affinities of STIC2 for Alb4C and Alb3C. In addition, it is possible that further Alb4C/STIC2 contact sites, which were not detected in our studies, contribute to binding. The higher affinity between Alb4C and STIC2, along with the finding that Alb3C but not Alb4C is capable of ribosome binding (Ackermann et al, 2021), suggests that the Alb4/STIC2 contact may support the correct repositioning of the RNC on the insertase machinery, while the C-terminal region of Alb3 anchors the ribosome directly. Our AlphaFold model depicts STIC2 as a dimer, with the Alb4 binding site positioned at the periphery of the dimer in the β-sheet region. Therefore, the internal partially hydrophobic cavity formed by the dimer remains unoccupied and may serve to bind other interaction partners.

Originally, stic2 has been identified along with alb4 (stic1) in a genetic screen for suppressors of the chlorotic tic40 A. thaliana mutant lacking the inner chloroplast envelope protein TIC40, which is involved in protein import (Bédard et al, 2017). The reason why the loss of STIC2 suppresses the phenotype of the tic40 mutation is not fully understood yet but indicates that STIC2 has an additional function in the posttranslational sorting of nuclear-encoded thylakoid membrane proteins as discussed by Bédard et al (2017), beyond its cotranslational role.

# Methods

### Reagents and tools table

| Reagent/Resource | Reference or Source | Identifier or Catalog Number |
|---|---|---|
| **Experimental Models** | | |
| Col-0 (*A. thaliana*) | ABRC | CS60000 |
| *stic2-3* (*A. thaliana*) | European Nottingham Arabidopsis Stock Center | SALK_001500 |
| *ffc1-2* (*A. thaliana*) | Amin et al, 1999 | N/A |
| *ffc1-2 stic2-3* (*A. thaliana*) | This study | N/A |
| Stellar (*E. coli*) | Takara | 636766 |
| DH5α (*E. coli*) | Thermo Fischer Scientific | EC0112 |
| XL1blue (*E. coli*) | Agilent | 200249 |
| Rosetta2 (DE3) pLysS (*E. coli*) | Novagen | 70956-M |
| BL21 (DE3) pLysS (*E. coli*) | Novagen | 69451-M |
| **Recombinant DNA** | | |
| pET-Duet1 | Novagen | 71146-3 |
| pET-Duet1-His-STIC2 (aa 49–182) | This study | N/A |
| pET29b | Novagen | 69872 |
| pET29b-Alb3C-His (aa 350–462) | Ackermann et al, 2021 | N/A |
| pET29b- Alb3CΔIII (aa 350–462 lacking aa 386–403) | This study | N/A |
| pET29b-Alb4C-His (aa 334–499) | Ackermann et al, 2021 | N/A |
| pET29b-Alb4CΔIII (aa 334–499 lacking aa 397–414) | This study | N/A |
| pET29b-Alb4C-His R339G | This study | N/A |
| pET29b-Alb4C-His F400G | This study | N/A |
| pET29b-Alb4C-His E407G | This study | N/A |
| pET29b-Alb4C-His R410G | This study | N/A |
| pGEX4T3 | Cytiva | 28954552 |
| pGEX4T3-GST-STIC2 | This study | N/A |
| pGEX4T3-GST-STIC2 G111L | This study | N/A |
| pGEX4T3-GST-STIC2 E112A | This study | N/A |
| pGEX4T3-GST-STIC2 K115A | This study | N/A |
| pGEX4T3-GST-STIC2 E131G | This study | N/A |
| pGEX4T3-GST-STIC2 E112A K115A | This study | N/A |
| **Antibodies** | | |
| Rabbit anti-cpSRP54M | Walter et al, 2015 | N/A |
| Rabbit anti-STIC2 | This study (Davids Biotechnology) | N/A |
| Rabbit anti-FLN1 | PhytoAB | PHY1889S |
| Rabbit anti-FZL | PhytoAB | PHY1890S |
| Rabbit anti-POR | Agrisera | AS05067 |
| Rabbit anti-TIG1 | Ries et al, 2017 | N/A |
| Rabbit anti-uL4 | Agrisera | AS153076 |
| Rabbit anti-uL24 | PhytoAB | PHY0426S |

| Reagent/Resource | Reference or Source | Identifier or Catalog Number |
|---|---|---|
| Rabbit anti-uS5 | Uniplastomic | AB003 |
| Rabbit anti-Twin-Strep-tag | IBA Liefscience | 2-1509-001 |
| Rabbit anti-SSU (Rbcs) | Agrisera | AS07259 |
| Rabbit anti-Alb3 | Bals et al, 2010 | N/A |
| Rabbit anti-Alb4 | Gerdes et al, 2006 | N/A |
| Rabbit anti-PsbA (D1) | Agrisera | AS05084 |
| Rabbit anti-PsbD (D2) | Agrisera | AS06146 |
| Rabbit anti-PsaA | Agrisera | AS06172 |
| Rabbit anti-PsaB | Agrisera | AS10695 |
| Mouse anti-Actin | Sigma-Aldrich | A0480 |
| Anti-His | Qiagen | 34460 |
| **Oligonucleotides and other sequence-based reagents** | | |
| cDNA STIC2 | Arabidopsis Biological Resource Center | U13069 |
| PCR primers | This study | Appendix Table S1 |
| **Chemicals, Enzymes and other reagents** | | |
| PRECISOR High-Fidelity DNA-polymerase | BioCat | N/A |
| Phusion High-Fidelity DNA Polymerase | Thermo Fischer Scientific | F530S |
| BamHI | Thermo Fischer Scientific | ER0051 |
| SalI | Thermo Fischer Scientific | ER0641 |
| T4 DNA ligase | Thermo Fischer Scientific | EL0014 |
| Percoll® | Cytiva | GE17-0891-01 |
| 10x BXT | IBA Lifescience | 2-1042-025 |
| [$^{35}$S]-methionine | Hartmann Analytics | SRM-01 |
| RiboLock RNase Inhibitor | Thermo Fischer Scientific | EO0382 |
| Nitrocellulose membrane | Macherey-Nagel | 741290 |
| PVDF membrane | Millipore | IPVH00010 |
| SuperSignal™ West Pico PLUS Chemiluminescent Substrate | Thermo Fischer Scientific | 34580 |
| SuperSignal West Femto substrate | Thermo Fischer Scientific | 34094 |
| Bio-Rad Protein Assay Dye Reagent Concentrate | BioRad | 5000006 |
| **Software** | | |
| ImageJ | https://imagej.net/software/imagej/ | N/A |
| Alphafold-Multimer | Evans et al, 2021; Jumper et al, 2021 | N/A |
| Colabfold 1.5.3 | Mirdita et al, 2022 | N/A |
| MaxQuant 1.6.10.43 | Cox and Mann, 2008 | N/A |
| **Other** | | |
| InFusion® HD EcoDry | Takara | 638955 |

| Reagent/Resource | Reference or Source | Identifier or Catalog Number |
|---|---|---|
| QuikChange Lightning Site-directed mutagenesis | Agilent Technologies | 210514 |
| HisTrap HP column, 5 ml | Cytiva | 17524802 |
| Glutathione Sepharose 4B GST-tagged protein purification resin | Cytiva | 17075601 |
| TranscriptAid T7 high yield transcription kit | Thermo Fischer Scientific | K0441 |
| MEGAclear™ transcription clean-up kit | Invitrogen | AM1908 |
| MagStrep® Strep-Tactin® XT beads | IBA Lifescience | 2-5090-002 |
| StrepTrap™ XT column, 1 ml | GE Healthcare | 29-4013-20 |

## Plant material and growth conditions

*A. thaliana* (Col 0) and *P. sativum* cv (Kelvedon Wonder) were grown on soil in an 8 h light/16 h dark cycle at a light intensity of 120 µmol photons m$^{-2}$ s$^{-1}$, a temperature of 22/19.5 °C (light/dark) and a relative humidity of 65%. For the Fv/Fm measurements, *A. thaliana* plants were grown in a 12 h light/12 h dark cycle at a light intensity of 50 µmol photons m$^{-2}$ s$^{-1}$, a temperature of 20/16 °C and a relative humidity of 60/75% (light/dark). For the light shift the light intensity was increased 10-fold from 50 to 500 µmol m$^{-2}$ s$^{-1}$, while maintaining all other parameters.

## Overexpression and purification of recombinant proteins

A cDNA clone coding for *A. thaliana* STIC2 (clone number U13069) was obtained from the Arabidopsis Biological Resource Center. For overexpression, the coding sequence for the mature STIC2 protein (aa 49–182; chloroplast transit peptide prediction according to TargetP) was cloned into BamHI/SalI linearized pETDuet1 (Novagen) and pGEX4T3 (Cytiva) using the InFusion® HD EcoDry™ kit (Takara). Primer used are indicated in Appendix Table S1. We later refer to mature STIC2 as STIC2.

Constructs for the expression of STIC2 containing different amino acid exchanges (STIC2 G111L, STIC2 E112A, STIC2 K115A, and STIC2 E112A K115A) were generated using the QuikChange Lightning Site-directed mutagenesis Kit (Agilent Technologies) with pET-Duet1-STIC2 as template. STIC2 E131G was generated via site-directed mutagenesis using 5′-phosphorylated primer as indicated in Appendix Table S1.

Cloning of the STIC2 site-directed mutagenesis constructs into the BamHI/SalI linearized pGEX4T3 vector was performed via restriction with BamHI and SalI (Thermo Fischer Scientific) and ligation using the T4-DNA ligase (Thermo Fischer Scientific) of the PCR products generated with the primer combinations as indicated in Appendix Table S1 and the corresponding pET-Duet1 constructs as template.

The pET29b (Novagen) constructs encoding the C-terminal region variants of Alb3 (Alb3C: aa 350–462) and Alb4 (Alb4C: aa 334–499) were described previously (Ackermann et al, 2021). The deletion of Motif III in Alb3C (Alb3CΔIII: aa 350–462 lacking aa 386–403) and Alb4C (Alb4CΔIII: aa 334–499 lacking aa 397–414)

were cloned into pET29b (Novagen) using the indicated primer (Appendix Table S1) as previously described (Ackermann et al, 2021).

Site-directed mutagenesis variants of Alb4C (R399G, F400G, E407G and R410G) were cloned using pET29b-Alb4C as template and the indicated primer (Appendix Table S1).

Recombinant proteins were expressed in *Escherichia coli* Rosetta2 (DE3) or BL21 (DE3) cells. The purification of His- or GST-tagged proteins was carried out with HisTrap HP prepacked columns or glutathione sepharose (Cytiva) under native conditions. After purification, His-tagged proteins were stored in 20 mM HEPES, 150 mM NaCl, pH 8.0. GST-tagged proteins were stored in 140 mM NaCl, 2.7 mM KCl, 10 mM Na$_2$HPO$_4$, 1.8 mM KH$_2$PO$_4$, pH 7.5 (PBS buffer).

## Preparation of translationally active S30 extract from pea chloroplasts

The isolation of intact chloroplasts from 8-day-old pea leaves was performed according to Walter et al (2015). The preparation of translationally active S30 extract was carried out as described (Yukawa et al, 2007).

## In vitro translation and RNC isolation

DNA templates containing a T7 promotor sequence and either full-length *psbA* from *P. sativum* including the 5′ untranslated region (UTR) from −87 to −1, or *rps2* from *A. thaliana* including the 5′ UTR from −90 to −1 were obtained from Invitrogen (GeneArt Gene Synthesis) and provided in the pMA vector backbone. In addition, the *psbA* and *rps2* DNA templates were designed to encode D1 and uS2c, respectively, with an internal Twin-Strep-tag® (TST) (IBA Lifesciences GmbH). The TST coding sequences were inserted at position +76 in *psbA* and position +52 in *rps2*. This results in a TST between amino acids E25/N26 and amino acids G17/V18 of the proteins D1 and uS2c, respectively. The TST coding sequence was additionally adapted to pea chloroplast codon usage according to the Kazusa codon usage database (www.kazusa.or.jp/codon/).

PCRs with PRECISOR High-Fidelity DNA-polymerase (BioCat) and various reverse primers (Appendix Table S1) lacking a stop codon were conducted to amplify DNA templates truncated from the 3′-end. The truncated PCR products were used for in vitro transcription using the TranscriptAid T7 high yield transcription kit (Thermo Scientific). Transcripts were purified with the MEGAclear™ transcription clean-up kit (Invitrogen).

In vitro translation was run for 60 min at 28 °C and illumination in 40 µl reactions (Walter et al, 2015; Yukawa et al, 2007). After addition of chloramphenicol (final concentration 0.85 mg/ml), RNCs were affinity-purified using the Strep-Tactin®XT System.

For mass spectrometric analyses, RNCs were isolated with MagStrep XT magnetic beads (IBA Lifescience). Therefore, 100 µl bead suspension was equilibrated in 30 mM HEPES pH 7.7, 200 mM NaCl, 70 mM potassium acetate, 9 mM magnesium acetate, 5 mM DTT containing 0.005% Tween20 (washing buffer) in a magnetic separator. A total of 160 µl translation reaction was incubated with equilibrated beads for 30 min at 4 °C with rotation. The supernatant was removed in the separator and the beads were washed twice with washing buffer and twice with washing buffer

without Tween20. The RNCs were eluted with BTX buffer (IBA Lifescience) (1xBTX, 70 mM potassium acetate, 10 mM magnesium acetate, 5 mM DTT) for 20 min (4 °C, rotation) and collected from the beads in the magnetic separator.

Isolation of RNCs for immunoblot analyses was performed using 1 ml prepacked StrepTrap XT columns and the ÄKTA-purifier (Cytiva). The columns were loaded with 320–880 µl translation reaction (0.1 ml/min) and washed (0.5 ml/min) with 10 column volumes washing buffer. Elution was done with 10 column volumes BTX buffer and collected in 0.5 ml fractions. Protein-containing fractions were then pooled, concentrated (Amicon Ultra-0.5, Merck), and subsequently equal amounts of protein according to OD$_{280}$ were applied to SDS-PAGE and immunoblot analyses.

## Mass spectrometry analyses

For tandem mass spectrometry (MS/MS), samples were prepared as described before (Kraus et al, 2020) using an in-gel tryptic digest. In brief, in vitro translation reactions were affinity-purified as described and the eluates were run on an SDS-PAGE until reaching the separation gel to remove impurities and cut out as a whole. The gel pieces of the different samples were washed and digested over night at 37 °C with MS-grade trypsin (Promega). MS/MS analysis was done using a previously described setup of a Waters nanoACQUITY gradient UPLC (www.waters.com) coupled to a Thermo Fisher Scientific Inc. Orbitrap ELITE mass spectrometer using a gradient over 60 min (Cormann et al, 2016; Kraus et al, 2020).

Data analysis was done with MaxQuant 1.6.10.43 (Cox and Mann, 2008) using a *P. sativum* database from 16.01.2019 of the "*Unité de Recherches en Génomique Info*" (URGI, UR1164, https://urgi.versailles.inra.fr/Species/Pisum) with additional annotations of candidates for potential homologs (Kreplak et al, 2019) (https://www.pulsedb.org/blast). In addition, the database was extended by the contaminant list of MaxQuant. For qualitative analysis, the Twin-Strep-tagged sequences of TST-D1 and TST-uS2c were added to the database. These were removed for quantitative analysis using label-free quantification (LFQ, Dataset EV1) due to the TST sequence redundancy. The precursor mass tolerance was set to 20 p.p.m. As modifications, methionine oxidation and N-terminal acetylation were included. The FDR of protein identification was set to 1%. A maximum of two missed cleavage sites was allowed.

Two independent preparations for each of TST-D1 RNC samples with different nascent peptide lengths (56–291) and the TST-uS2c (158) RNC sample, respectively, were analyzed in technical duplicates by MS/MS. For the evaluation of the MS/MS data, for each of the TST-D1 RNC and TST-uS2c RNC samples only protein IDs with LFQ intensities in both biological replicates were included. For the included protein IDs, LFQ intensity means for each biological replicate were calculated from the corresponding technical duplicate values. The protein IDs were then categorized as D1/uS2c overlap, D1 or uS2c exclusive based on their detection in the TST-D1 and TST-uS2 RNC samples. This was done either for each TST-D1 chain length individually, or for those grouped together as short (56 and 69), medium (108 and 136) and long (195 and 291) D1 chains. These groups are based on the exposure of transmembrane helices (TMHs) of the nascent D1 peptide, with the first TMH still fully buried inside the ribosome peptide tunnel for

*short* D1 RNCs, an exposed first TMH in *medium* D1 RNCs, and at least two exposed TMHs in *long* D1 RNCs. Based on the LFQ intensities of the biological replicates, a statistically significant enrichment of protein groups in the D1/uS2c overlap category was determined by a two-sided unpaired *t*-test ($p$-value ≤ 0.05), and log$_2$ fold-change values of D1 over uS2c were calculated, in both cases for the individual and grouped TST-D1 chains. Volcano plots are used to depict the LFQ fold-change (log$_2$) of D1 over uS2c against $p$-values ($-$log$_{10}$). A gray line represents the $p$-value threshold (≤0.05) (Appendix Fig. S2).

## Measurement of the maximum quantum yield of PSII

The maximum quantum yield of PSII (Fv/Fm) was determined using the IMAGING-PAM system (Walz GmbH, Germany) with a prior dark acclimation of plants for 30 min. To analyze differences due to leaf age the three youngest fully developed leaves (3–5) were defined as young and leaves 9–11 as mature, respectively. The Games-Howell multiple comparison test, which allows the analysis of data with unequal variances, was used to test for significant differences between genotypes.

## Total protein extraction of *A. thaliana* leaves

Total protein extracts from 5-week-old *A. thaliana* leaves were prepared as described previously (Hristou et al, 2019).

## Isolation and fractionation of *A. thaliana* chloroplasts

Chloroplast isolation and fractionation was previously described (Hristou et al, 2019). Gels for SDS-PAGE were loaded on a proportional chlorophyll basis (5 µg) with whole chloroplasts, stroma and thylakoids and subsequently applied to Coomassie® Brilliant Blue R-250 staining or immunoblot analyses.

## In vivo labeling and pulse-chase experiments

In vivo labeling and pulse-chase experiments were performed as described previously with the following modifications (Bonardi et al, 2005). For radioactive labeling of chloroplast proteins, leaf discs (4 mm diameter) of 4- to 5-week-old *A. thaliana* plants were vacuum infiltrated in 1 mM KH$_2$PO$_4$ pH 6.3, 0.1% (v/v) Tween 20, 0.1 µCi/µl [$^{35}$S]-methionine (>1000 Ci/mmol, Hartmann Analytics) and 20 µg/ml cycloheximide (labeling buffer) for the indicated time periods in ambient light at room temperature. To isolate thylakoid membranes, two leaf discs were homogenized with a conical stainless-steel rod. The homogenate was centrifuged (10 min, 16,000 × $g$, 4 °C) and the membrane pellet was washed with 50 µl of 1 mM KH$_2$PO$_4$ pH 6.3, 0.1% (v/v) Tween 20. The membranes were finally resuspended in 25 µl sample buffer.

To monitor D1 degradation, chloroplast proteins were pulse labeled as described above for one hour at ambient light and room temperature. After washing off remaining labeling buffer, leaf discs were briefly (10 min) vacuum-infiltrated in 1 mM KH$_2$PO$_4$ pH 6.3, 0.1% (v/v) Tween 20, 10 mM L-methionine and 20 µg/ml cycloheximide (chase buffer) followed by an incubation period of 5 h under high light (1000 µmol photons m$^{-2}$ s$^{-1}$). After every hour thylakoids of two leaf discs were isolated and applied to SDS-PAGE.

## Ribosome purification

400 mg leaf material of 5-week-old *A. thaliana* plants was harvested during the light cycle and immediately frozen in liquid nitrogen. The leaf material was pulverized in liquid nitrogen and homogenization was carried out in 1 ml extraction buffer (200 mM Tris-HC pH 9.0, 200 mM KCl, 35 mM MgCl$_2$, 25 mM EGTA, 200 mM sucrose, 0.5 mg/ml heparin, 10 mM β-mercaptoethanol, 100 μg/ml chloramphenicol, 100 μg/ml cycloheximide, 1 mM phenylmethylsulfonylfluoride, 1% (v/v) Triton X-100, 2% (v/v) polyoxyethylene tridecyl ether). 0.5 U/ml RiboLock RNase Inhibitor (Thermo-Scientific) were added. After 10 min incubation on ice the sample was centrifuged (16,000 × g, 10 min, 4 °C). The supernatant was mixed with 0.5% (w/v) sodium deoxycholate and centrifuged (12,000 × g, 15 min, 4 °C). 400 μl of the supernatant were loaded on a sucrose gradient (15–40%). The gradients were centrifuged (175,000 × g, 17 h, 4 °C, acceleration/deceleration 5) in the Optima™ Max-XP Tabletop Ultracentrifuge (Beckmann) using the MLS-50 Swinging-Bucket rotor. Gradients were fractionated in 18 fractions and 10 μl of each fraction were separated via SDS-PAGE and analyzed by immunoblotting.

## Ribosome profiling

The spatial examination of stromal and thylakoid membrane-associated ribosomes of the 3-week-old wild-type and mutant plants was performed as described previously (Hristou et al, 2019) with one minor modification: after separation of membrane and soluble ribosomes, footprints were generated with 600 U RNaseI/ml lysate.

## Pepspot analyses

Peptide libraries were obtained from JPT Peptide Technologies GmbH, Berlin. For Alb3, peptide membranes with 11 and 43 peptide spots covering amino acids 155–208 and 282–462, respectively, were designed (15mer peptides, overlapping by 11 amino acids). Alb4 peptide membranes comprised amino acids 139–192 (11 spots) and 266–499 (56 spots) (15mer peptides, overlap of 12 amino acids).

According to manufactures instructions, the membranes were first incubated in methanol and then washed with 50 mM Tris-HCl, 150 mM NaCl, 0.3% (v/v) Tween 20, pH 7.5 (TBST buffer). Blocking of the membranes was performed in anti-His HRP conjugate blocking buffer (Qiagen). After blocking at room temperature for 2 h, His-STIC2 (5 μg/ml) was added and incubated for another 3 h. After washing with TBST, bound His-STIC2 protein was detected by anti-His HRP conjugate (Qiagen) and enhanced chemiluminescence reaction (Pierce).

## Isothermal titration calorimetry (ITC)

The experiments were carried out in 20 mM HEPES, 150 mM NaCl, pH 8.0 (ITC buffer) at 25 °C using an Auto-iTC200 (MicroCal) (GE Healthcare). 1.0 mM and 1.5 mM mature STIC2 was titrated into solutions of 0.1 mM Alb4C or Alb4CΔIII and 0.15 mM Alb3C or Alb3CΔIII, respectively. Control experiments were performed by titration of STIC2 solution into ITC buffer. Data analysis and fitting was carried out with help of the manufacturer´s software using a one-site binding model.

## Pulldown analyses

For pulldown experiments, 20 μg of GST-STIC2 variants were incubated with 20 μg of Alb4C-His variants and Alb3C-His in PBS and PBS with 160 mM NaCl, respectively, in a final volume of 120 μl (10 min, room temperature). GST-eGFP was used as a control. A 10 μl load sample was taken and mixed with sample buffer. Subsequently, 100 μl of equilibrated glutathione sepharose (Cytiva) was added to the pulldown reaction and incubated for another 30 min (rotating, room temperature). Next, the pulldown reaction was loaded on a Wizard® Minicolumn (Promega) and the supernatant was removed by short spinning. Unbound proteins were washed out with 5 ml PBS or PBS with 160 mM NaCl and the remaining supernatant was removed by centrifugation (17,000 × g, 30 s). For elution, 30 μl PBS with 160 mM NaCl and 10 mM reduced L-glutathione was added to the column and incubated for 15 min. Eluted proteins were collected by centrifugation (17,000 × g, 30 s) and mixed with 15 μl sample buffer. Subsequently, 15 μl of load and 10 μl of eluate samples were separated on SDS-PAGE. Load samples were stained with Coomassie® *Brilliant Blue* R-250, eluates were subsequently analyzed via immunoblotting.

## Immunoblot analyses

Proteins separated by SDS-PAGE were blotted onto nitrocellulose (Macherey-Nagel) or PVDF membranes (Millipore). Proteins were immunodetected by specific antibodies against cpSRP54M (Walter et al, 2015), STIC2 (produced in rabbit immunized with recombinant His-STIC2 (Davids Biotechnology)), FLN1 (PhytoAB, PHY1889S), FZL (PhytoAB, PHY1890S), POR (Agrisera, AS05067), TIG1 (kind gift from F. Willmund (Ries et al, 2017)), uL4 (Agrisera, AS153076), uL24 (PhytoAB, PHY0426S), uS5 (Uniplastomic, AB003), Twin-Strep-tag (IBA Lifescience, 2-1509-001), Rbcs (SSU) (Agrisera, AS07259), Alb3 (Bals et al, 2010), PsbA (D1) (Agrisera, AS05084), PsbD (D2) (Agrisera, AS06146), PsaA (Agrisera, AS06172), PsaB (Agrisera, AS10695), Actin (Sigma-Aldrich, A0480), the His$_6$-tag (Quiagen, 34460), Alb4C (Gerdes et al, 2006) and enhanced chemiluminescence reaction (Thermo Fisher Scientific).

## Modeling

STIC2 and Alb4C were submitted to Alphafold-Multimer (Evans et al, 2021; Jumper et al, 2021) prediction by using the Localcolabfold pipeline (Colabfold 1.5.3, Mirdita et al, 2022). Several combinations were tested, but most consistent results were obtained submitting two copies of each protein to form a heterotetramer. Colabfold was used with the options to use templates and to relax the final models using the Amber forcefield, resulting in five independent relaxed and ranked complex models. While the overall prediction quality for STIC2 was relatively high as indicated by pLDDT values over 90 for most of the core region, for Alb4C only a central elongated helix was predicted with high confidence (Appendix Fig. S6A). All five models superposed nearly perfectly in the regions discussed in the text, and showed perfect twofold symmetry along the long STIC2-axis. For figure preparation, the highest-ranking complex model was used, focusing on the interface between chains A of STIC2 and chain C of Alb4C. The same procedure was applied to Alb3C, with very similar outcome (Appendix Fig. S6A).

## Data availability

The mass spectrometry proteomics data have been deposited to the ProteomeXchange Consortium (http://proteomecentral.proteomexchange.org) via the PRIDE partner repository (Perez-Riverol et al, 2022) with the dataset identifier PXD042896.

The source data of this paper are collected in the following database record: biostudies:S-SCDT-10_1038-S44318-024-00211-4.

## Peer review information

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

## Acknowledgements

We like to acknowledge Silke Funke for technical assistance and Juliane Walter as well as Julia Bollmann for help with the experiments. We thank Felix Willmund for the kind gift of antibodies against TIG1. This work was supported by the Deutsche Forschungsgemeinschaft (DFG) (SCHU 1163/6-2 and 836/3-2 within research unit FOR2092 to DS and MMN, respectively; BA 1902/2-2 to SB). UA received funding from the Max Planck Society (MPG). RZ is funded by the DFG (ZO 302/5-1) and the Max Planck Society (MPG).

## Author contributions

**Dominique S Stolle**: Conceptualization; Data curation; Investigation; Visualization; Methodology; Writing—original draft. **Lena Osterhoff**: Conceptualization; Data curation; Investigation; Visualization; Methodology; Writing—review and editing. **Paul Treimer**: Investigation. **Jan Lambertz**: Resources; Data curation; Supervision; Investigation; Visualization; Writing—review and editing. **Marie Karstens**: Investigation; Visualization. **Jakob-Maximilian Keller**: Investigation; Visualization. **Ines Gerlach**: Investigation. **Annika Bischoff**: Data curation; Investigation; Visualization; Writing—review and editing. **Beatrix Dünschede**: Investigation; Visualization. **Anja Rödiger**: Data curation; Investigation. **Christian Herrmann**: Resources; Data curation; Supervision. **Sacha Baginsky**: Resources; Data curation; Supervision. **Eckhard Hofmann**: Conceptualization; Resources; Software; Investigation; Visualization; Writing—review and editing. **Reimo Zoschke**: Conceptualization; Resources; Data curation; Supervision; Visualization; Writing—review and editing. **Ute Armbruster**: Conceptualization; Resources; Data curation; Supervision; Writing—review and editing. **Marc M Nowaczyk**: Conceptualization; Resources; Data curation; Supervision; Writing—review and editing. **Danja Schünemann**: Conceptualization; Resources; Data curation; Formal analysis; Supervision; Funding acquisition; Validation; Writing—original draft; Project administration; Writing—review and editing.

Source data underlying figure panels in this paper may have individual authorship assigned. Where available, figure panel/source data authorship is listed in the following database record: biostudies:S-SCDT-10_1038-S44318-024-00211-4.

## Funding

## Disclosure and competing interests statement

The authors declare no competing interests.

