## [Peer Review File · The EMBO Journal]

STIC2 selectively binds ribosome-nascent chain complexes in the cotranslational sorting of *Arabidopsis* thylakoid proteins

Danja Schunemann, Dominique Stolle, Lena Osterhoff, Paul Treimer, Jan Lambertz, Marie Karstens, Jakob-Maximilian Keller, Ines Gerlach, Annika Bischoff, Beatrix Dünschede, Anja Rödiger, Christian Herrmann, Sacha Baginsky, Eckhard Hofmann, Reimo Zoschke, Ute Armbruster, and Marc Nowaczyk

Corresponding author(s): Danja Schunemann (danja.schuenemann@rub.de)

Review Timeline:

Submission Date:	22nd Jun 23
Editorial Decision:	7th Aug 23
Revision Received:	14th Jun 24
Editorial Decision:	9th Jul 24
Revision Received:	24th Jul 24
Accepted:	26th Jul 24

Editor: William Teale

Transaction Report:

Dear Danja,

Thank you for submitting your manuscript to The EMBO Journal. We have now received comments from three reviewers, which are included below for your information.

As you will see from the reports, the reviewers find the study interesting, but they also raise multiple substantial concerns, in particular regarding the novelty of your work when your manuscript is compared to the study of Bedard et al., 2017 (28684427). All referees ask for further mechanistic insights into STIC2 activity, which may follow in the direction indicated by point 3 of referee #3. The biochemical relationship between STIC2 and cpSRP54 may also be explored.

I realise that this would call for a rather extensive revision, which might not be feasible within our standard 3-6 month time frame. Therefore, I would appreciate your could taking a look at the reviewer comments and let me know to what extent you would be prepared to address the raised issues in a revision by providing a preliminary point-by-point response. This would be very helpful for the final editorial decision.

I could also at this stage (if you would like) approach the editorial team at EMBO Reports on your behalf and ask about which points would have to be addressed for publication there.

Please feel free to contact me if you have any questions regarding this pre-decision consultation approach. I look forward to your response.

Best wishes,

William

William Teale, PhD
Editor
The EMBO Journal
w.teale@embojournal.org

Please remember: Digital image enhancement is acceptable practice, as long as it accurately represents the original data and conforms to community standards. If a figure has been subjected to significant electronic manipulation, this must be noted in the figure legend or in the 'Materials and Methods' section. The editors reserve the right to request original versions of figures and

the original images that were used to assemble the figure.

We realize that it is difficult to revise to a specific deadline. In the interest of protecting the conceptual advance provided by the work, we recommend a revision within 3 months (5th Nov 2023). Please discuss the revision progress ahead of this time with the editor if you require more time to complete the revisions. Use the link below to submit your revision:

Referee #1:

In this article, Stolle et al. report on the identification of proteins interacting with chloroplast ribosomes during the co-translational insertion of the D1 protein into thylakoid membranes. For this, the authors used their previously established in vitro translation system based on pea chloroplast lysates with truncated psbA and μ S2c transcripts containing the coding sequence for a Twin-Strep-tag. Stranlation complexes were then isolated using streptavidin columns and analyzed by mass spectrometry. The authors found 119 proteins in all samples (among them as expected 34 ribosomal subunits) and 123 proteins specifically enriched in samples from stalled D1 translation complexes. Here differential association depending on the length of the translated D1 protein was discovered. The list of proteins makes much sense and provides an entry point for many future studies on the mechanisms underlying cotranslational insertion of thylakoid membrane proteins. The authors started this by focussing on cpSRP54 and STIC2, which were specifically enriched with all D1 translation products versus the μ S2c control. They show that STIC2 interacts with stromal ribosomes and with motif III of the ALB3 and ALB4 insertases. Furthermore, the authors demonstrate that the absence of stic2 severely aggravates the PSII phenotype in the cpSRP54 mutant ffc1, specifically leading to reduced synthesis of D1 and reduced accumulation of D1 and PsaB in young leaves. The article is very well written, the data are nicely presented and will be an invaluable resource for the community. Publication in a high-impact journal is fully justified. Nevertheless, the outcome of some experiments would be much more convincing if they were based on quantification of replicates, as detailed below. Moreover, I would like to have some critical discussion on potential caveats of this innovative approach.

Major points:

- The MS data are based on two biological replicates which have been measured twice (technical replicates). I assume that the statistical tests were performed based on the two biological replicates after averaging the technical ones. This needs to be stated somewhere. It would not be acceptable if the two biological and two technical replicates were treated as four biological replicates in the statistical tests. Please also indicate which correction was used for multiple testing.
- I wonder whether the Strep tag could interfere with proper folding of the translated polypeptide. Could this be the reason for the interaction with post-translationally acting chaperones like CPN60, HSP90, CLPB3 and the proteases? Would the full-length proteins be functional with the Strep tag sequence included at the positions chosen? Some discussion of this, e.g. after line 438, would be strongly appreciated.
- While the finding of the many enzymes involved in metabolism is in line with earlier results, I still wonder if this could be due to the artificial situation in the in vitro translation system. Here membrane-less compartmentalization is likely disturbed putting proteins into proximity that are not in proximity in vivo. This cannot be avoided but some discussion on this aspect would be desirable (line 386).
- Figure 4B is convincing for the ALB3 C-terminal deletion but not for that of ALB4. Please quantify the ratio of pulled down ALB3/4 (WT and deletion) to the respective STIC2 bait. This should be done in replicates, allowing for statistical tests (which should be shown).
- Line 330: it appears as if the stic2 mutant has only 50% of D1, while the authors claim that levels are similar. Please quantify immunoblot data shown in Fig.5B (including replicates).
- Figure 5C, D: please quantify in replicates, make statistical tests; if possible, include signals for other proteins as controls.

Minor points:

- Supplemental Data sets: please add the number of unique peptides identified.
- Line 177: classified as biogenesis factors based on what?
- Line 249: indicate the fractions you are referring to.
- Figure S6: it seems as if less RC was accumulating in the ffc mutant, none in the stic2/ffc double mutant. If so, please add this information.
- Line 360: avoid "for the first time" statements, at the end a similar experiment was already done that is hidden in the literature
- Is there data available from pulse experiments on PsaA/B synthesis rates? Could they be used to put flesh on the speculation in line 465?

Referee #2:

Chloroplast contains over 3000 proteins, of which 90% are encoded by the nuclear genome, translated in the cytosol and imported into chloroplast in a post-translational manner. However, there are still a small population of proteins encoded by the plastid genome and synthesized inside chloroplast. Most of them are essential components of photosynthetic complexes on the thylakoid membrane, such as D1, D2 and PetB. These proteins usually are synthesized in nascent chains, and then inserted into the membrane by the co-translational mechanism through the cpSRP pathway. This process is elegantly orchestrated by an extensive network of factors that help membrane targeting, insertion, folding and maturation of the nascent chain. The homologous system in bacteria has been investigated more thoroughly and the mechanism is much better understood. However, despite of its prokaryotic origin, chloroplast seems to evolve its own mechanism for membrane protein biogenesis. Targeting of these proteins is believed to be initiated upon direct interaction of cpSRP54 with the ribosomal subunit uL4. Then the nascent chains are delivered to the cpSecY translocon channel. After that, the Alb3 integrase facilitates release of the chains from the translocon and incorporation into membrane environment. However, the mechanism of how nascent chains are distinguished from the nuclear-encoded polypeptides by cpSRP54 and delivered to the translocon in the membrane is still unknown.

In this study, Stolle et al., developed an approach by generating a stalled ribosome-nascent chain complex bearing the translating membrane D1 protein and performing affinity purification in tandem with mass spectrometry. They identified a list of candidate proteins that associate with the stalling ribosomes, including some well-established factors in the cpSRP pathway. Interestingly, STIC2, which was previously identified as a stroma protein involved in thylakoid membrane biogenesis is shown to play a role in D1 protein targeting. They proposed that STIC2 functions in the pathway by interacting with both ribosomes and the insertase Alb3 and Alb4. These findings are interesting and some of the data are of high quality. However, because most results have been reported in a previous study (Bedard et al, 2017), the novelty of this study is not appreciated. Furthermore, the reviewer did not see too much mechanical insights of STIC2 to explain its role in the cpSRP pathway beyond the Bedard's paper. Based on the current data, it's very hard to reach the conclusion that STIC2 is a specific factor which only delivers the co-translational substrates in the cpSRP pathway. Here are some major concerns:

1. The data supporting STIC2 association with ribosomes is rather weak. It looks to me that the co-migration of STIC2 with ribosomes on the size exclusion chromatography is not convincing. The author argue that prepared ribosomes are largely translationally inactive or predominantly translate soluble proteins. It would be nice to see the better comigration of STIC2 and translating ribosomes on the thylakoid membrane if the membrane fraction is isolated for immunoblot.
2. In the pull-down assay, STIC2 interacts much stronger with alb3 than with alb4. However, opposite conclusion was obtained from the ITC experiment. This discrepancy needs to be explained or tested by other experiments.
3. It's difficult to understand that stic2 knockout mutant has no phenotype under any condition, given its role in D1 protein biogenesis. There are a few factors, like LPA1 or PAM68 have been reported to be involved in D1 protein assembly into the photosystem complex. Most of the mutants lacking these factors display severe phenotype of either growth or photosynthesis. Based on the proposed model, I expect to see some defects of the stic2 knockout mutant. The synergistic effect of STIC2 and cpSRP54 is obvious but also raises the question of how much percent of the nascent chains truly goes through STIC2. Does STIC2 interact with cpSRP54, or do they work independently? It's possible that STIC2 only mediates a very small number of D1 protein insertion and cpSRP54 plays more predominant roles. Can STIC2 overexpression rescue the phenotype of the ffc mutant at some level? In any way, the mechanism of STIC2 is not well clarified and this point should be the highlight of this study.

Minor point:

The high background of the pulldown experiment shown in Fig S4 need to be addressed. There are still quite lots of protein bound non-specifically in the negative control. It might be worth to swap the bait and prey to repeat the experiment.

Referee #3:

In this manuscript, Stolle et al. report on the use of affinity-purification-based label-free quantitative proteomics, together with cell biology, biochemical and biophysical analyses and photosynthetic parameter measurements, to identify a D1 RNC interactome and then to point to a role of the STIC2 protein in D1 targeting and biogenesis. The manuscript includes a wealth of high-quality data that is nicely presented and so is very informative. This important study will be of considerable interest to those in the relevant intersecting fields of organelle biology, protein biogenesis, and photosynthesis. While the D1 RNC interactome is a significant dataset in its own right, the manuscript would be considerably strengthened if additional data could be added that provide significant new insights into how the STIC2 protein exerts its molecular function (i.e. mechanism) in D1 targeting and PSII biogenesis. Some specific comments are below:

1. As the authors pointed out in the Discussion, a direct interaction of STIC2 with both thylakoid membrane proteins, Alb3 and Alb4, had been reported before (Figure 6 in Bedard et al., 2017). In vitro pulldown experiments were used in both studies (e.g. Alb3 aa 350-462 and aa 361-462 were used here and in Bedard et al., 2017, respectively; Alb4 aa 334-499 and aa 345-498 were used here and in Bedard et al., 2017, respectively), and the same observations and conclusions were made (i.e. the conserved C-terminal motif III of Alb3 and Alb4 interacts with STIC2). It would greatly strengthen this manuscript if the data here could be extended to take it beyond what was previously reported.

2. The same genetic analysis (i.e. the single and double mutants of *stic2* and *ffc1-2= cpsrp54*) was conducted here and by Bedard et al., 2017. Although different methods (e.g. Fv/Fm measurements, immunoblotting, and pulse-chase assay) were used here to compare the phenotype between the different genetic backgrounds, this new analysis does not provide much additional insight into how STIC2 exerts its functions in D1 synthesis or STIC2's relationship with the cpSRP54 pathway. Again, it would add considerable value to the manuscript if significant new insight could be added here.

3. One addition that would considerably add to the novelty/conceptual advance here would be the identification and characterization of the binding site between STIC2 and the D1 RNC, and how STIC2 is recruited the D1 RNC.

4. Fold-change of proteins with significant quantitative differences (i.e. enriched) needs to be presented in the tables and supplemental datasets.

5. Figure 5C. A full gel of the pulse-labelling assay is required to see whether the observed effect is D1 specific or is due to a potential methionine uptake issue in the double mutant. Quantitative analysis of these data (C and D), with replicates and statistical analysis, is also needed.

6. Line 456. Alb4 also contains motif II according to Trösch et al. (2015b).

7. Lines 470-474. Here, it would be helpful if the authors would comment on the hypotheses proposed by Bedard et al. (2017).

Point-by-point responses to the Reviewer:

First, I would like to thank the Reviewers for their critical evaluation of our manuscript. Your feedback was very helpful to improve the quality and impact of our work. Below, you will find our point-by-point responses to your comments.

Additionally, we attached a revised version of the manuscript with all changes in the main text highlighted, except for minor wording improvements, which were not highlighted. This highlighted version follows the manuscript without marked sections. We also changed the sequence of the Figures and included a new Figure 6. Furthermore, we slightly adapted the manuscript's title.

Referee #1:

In this article, Stolle et al. report on the identification of proteins interacting with chloroplast ribosomes during the co-translational insertion of the D1 protein into thylakoid membranes. For this, the authors used their previously established in vitro translation system based on pea chloroplast lysates with truncated psbA and μ S2c transcripts containing the coding sequence for a Twin-Strep-tag. Stranlation complexes were then isolated using streptavidin columns and analyzed by mass spectrometry. The authors found 119 proteins in all samples (among them as expected 34 ribosomal subunits) and 123 proteins specifically enriched in samples from stalled D1 translation complexes. Here differential association depending on the length of the translated D1 protein was discovered. The list of proteins makes much sense and provides an entry point for many future studies on the mechanisms underlying cotranslational insertion of thylakoid membrane proteins. The authors started this by focussing on cpSRP54 and STIC2, which were specifically enriched with all D1 translation products versus the μ S2c control. They show that STIC2 interacts with stromal ribosomes and with motif III of the ALB3 and ALB4 insertases. Furthermore, the authors demonstrate that the absence of stic2 severely aggravates the PSII phenotype in the cpSRP54 mutant ffc1, specifically leading to reduced synthesis of D1 and reduced accumulation of D1 and PsaB in young leaves. The article is very well written, the data are nicely presented and will be an invaluable resource for the community. Publication in a high-impact journal is fully justified. Nevertheless, the outcome of some experiments would be much more convincing if they were based on quantification of replicates, as detailed below. Moreover, I would like to have some critical discussion on potential caveats of this innovative approach.

Major points:

- The MS data are based on two biological replicates which have been measured twice (technical replicates). I assume that the statistical tests were performed based on the two biological replicates after averaging the technical ones. This needs to be stated somewhere. It would not be acceptable if the two biological and two technical replicates were treated as four biological replicates in the statistical tests. Please also indicate which correction was used for multiple testing.

Answer:

Thank you for your valuable feedback regarding the statistical analysis of the MS data. In the initial version of the manuscript, the statistical analysis of the MS data treated biological and technical replicates independently. We agree with the reviewer that the statistical analysis is more appropriate if only the two biological replicates, averaged over the technical replicates, are included. Therefore, we have recalculated the statistics of our mass spectrometry. This resulted in certain changes, which, however, do not have impact on the main results of the manuscript. We highlighted all changes related to this recalculation in the main text of the revised version of the manuscript. We adapted Figure 2 A/B, Tables 2 and 3, the Datasets and the data shown in the Appendix to the new calculation. I would like to note that Table 1, which lists the identified biogenesis factors (cpSRP54, STIC2, FZL, FLN1) remains unchanged. The Materials and Methods section was adapted and includes a detailed description of the MS data analysis.

- I wonder whether the Strep tag could interfere with proper folding of the translated polypeptide. Could this be the reason for the interaction with post-translationally acting chaperones like CPN60, HSP90, CLPB3 and the proteases? Would the full-length proteins be functional with the Strep tag sequence included at the positions chosen? Some discussion of this, e.g. after line 438, would be strongly appreciated.

Answer:

Yes, this is a good suggestion. In the revised manuscript we added a discussion about the possible effects of the Strep tag (lines 519-521).

- While the finding of the many enzymes involved in metabolism is in line with earlier results, I still wonder if this could be due to the artificial situation in the in vitro translation system. Here membrane-less

compartmentalization is likely disturbed putting proteins into proximity that are not in proximity in vivo. This cannot be avoided but some discussion on this aspect would be desirable (line 386).

Answer:

We agree and discuss this point in the revised version (lines 463-467).

- Figure 4B is convincing for the ALB3 C-terminal deletion but not for that of ALB4. Please quantify the ratio of pulled down ALB3/4 (WT and deletion) to the respective STIC2 bait. This should be done in replicates, allowing for statistical tests (which should be shown).

Answer:

Thank you for your feedback. We have significantly improved the manuscript and replaced the original Figure 4B with a new Figure 6. Figure 6 describes the characterization of the binding interface between the C-terminal region of Alb3/Alb4 and STIC2. The new data confirm our pepspot and ITC data that Motif III is crucial for binding and additionally identify the β -sheet region of STIC2 as docking site. The new experiments include pulldown experiments using site directed mutagenesis constructs of both interaction partners. The new Fig. 6 includes a quantification of the pull-down assays.

- Line 330: it appears as if the *stic2* mutant has only 50% of D1, while the authors claim that levels are similar. Please quantify immunoblot data shown in Fig.5B (including replicates).

Answer:

We quantified the immunoblot data and expanded the analysis to include all reaction center subunits of PSII (D1 and D2) and PSI (PsaA and PsaB). This data set is shown in the revised manuscript as Figure 3B. We observed a slight decrease of all these photosynthetic proteins in the *stic2* mutant. This additional information is now included in the revised version (lines 285-286). We appreciate the reviewer's valuable suggestion. The new data are important as they indicate that the *stic2* single mutant is slightly impaired in photosystem biogenesis.

- Figure 5C, D: please quantify in replicates, make statistical tests; if possible, include signals for other proteins as controls.

Answer:

We repeated the pulse experiments and included the *stic2* single mutant. The revised figure (now Fig. 3C) includes the quantification of the results. Since the pulse-chase experiments showed no significant differences in the D1 degradation, we decided not to repeat and further quantify these experiments. In the revised manuscript, we present the degradation assay as Appendix Figure S4A, showing one representative experiment from three independent biological replicates.

In the labeling experiments, D1 was the prominent signal detected. We also observed some additional weak signals that could not be confidently assigned to specific proteins, though they showed reduced synthesis in the *ffc stic2* double mutant and might correspond to PsaA or PsaB (see also my response to the point below). Therefore, it is not feasible to include a signal from an unrelated equally labeled protein as a control. Additionally, using a Coomassie-stained gel for loading controls between mutants is also not suitable due to the significant differences in protein patterns.

Minor points:

- Supplemental Data sets: please add the number of unique peptides identified.

Answer:

We added the number of unique peptides and also the number of proteins and peptides to allow a more precise overview of our data in the Dataset EV4 (previously Dataset S4). Please note that due to the quality of the reference database, which may contain duplicate or fragment entries, the values for peptides and unique peptides differ for a few proteins.

- Line 177: classified as biogenesis factors based on what?

Answer:

The classification is based on the section "Function" of the UniProt database and on current literature.

- Line 249: indicate the fractions you are referring to.

Answer:

In the first version of the manuscript line 249 refers to Figure 3A. In the revised manuscript Figure 3A was replaced with a new experiment (now Figure 4A). Accordingly, the main text was rewritten. The new experiment shows a clear cofractionation of STIC2 with ribosomes to the bottom fraction of a sucrose gradient.

- Figure S6: it seems as if less RC was accumulating in the *ffc* mutant, none in the *stic2/ffc* double mutant. If so, please add this information.

Answer:

We acknowledge the reviewer's observation regarding the differences in RC signal accumulation as depicted in the BN-PAGE results. However, this variability was not consistently observed across our replicates. The RC signal was generally faint in all our samples and exhibited some degree of variability, even in the wild-type samples.

- Line 360: avoid "for the first time" statements, at the end a similar experiment was already done that is hidden in the literature

Answer:

While we believe that the previously described experiments are quite different from our study, we revised our text according to the reviewer's concern (line 437).

- Is there data available from pulse experiments on *PsaA/B* synthesis rates? Could they be used to put flesh on the speculation in line 465?

Answer:

Due to the low turnover rate of *PsaA/B*, it is challenging to detect these proteins in pulse-labeling experiments. In our labeling experiments, D1 was the only protein that was clearly labeled and could be unambiguously identified. We observed two faintly labeled bands in the 55-70 kDa range, whose synthesis was also reduced in *ffc1-2 stic2-3*, and which may represent *PsaA/B*. However, because these signals were very weak and we could not confidently assign these bands, we decided not to include these data to avoid the risk of misinterpretation.

Referee #2:

Chloroplast contains over 3000 proteins, of which 90% are encoded by the nuclear genome, translated in the cytosol and imported into chloroplast in a post-translational manner. However, there are still a small population of proteins encoded by the plastid genome and synthesized inside chloroplast. Most of them are essential components of photosynthetic complexes on the thylakoid membrane, such as D1, D2 and PetB. These proteins usually are synthesized in nascent chains, and then inserted into the membrane by the co-translational mechanism through the cpSRP pathway. This process is elegantly orchestrated by an extensive network of factors that help membrane targeting, insertion, folding and maturation of the nascent chain. The homologous system in bacteria has been investigated more thoroughly and the mechanism is much better understood. However, despite of its prokaryotic origin, chloroplast seems to evolve its own mechanism for membrane protein biogenesis. Targeting of these proteins is believed to be initiated upon direct interaction of cpSRP54 with the ribosomal subunit uL4. Then the nascent chains are delivered to the cpSecY translocon channel. After that, the Alb3 integrase facilitates release of the chains from the translocon and incorporation into membrane environment. However, the mechanism of how nascent chains are distinguished from the nuclear-encoded polypeptides by cpSRP54 and delivered to the translocon in the membrane is still unknown.

In this study, Stolle et al., developed an approach by generating a stalled ribosome-nascent chain complex bearing the translating membrane D1 protein and performing affinity purification in tandem with mass spectrometry. They identified a list of candidate proteins that associate with the stalling ribosomes, including some well-established factors in the cpSRP pathway. Interestingly, STIC2, which was previously identified as a stroma protein involved in thylakoid membrane biogenesis is shown to play a role in D1 protein targeting. They proposed that STIC2 functions in the pathway by interacting with both ribosomes and the insertase Alb3 and Alb4. These findings are interesting and some of the data are of high quality. However, because most results have been reported in a previous study (Bedard et al, 2017), the novelty of this study is not appreciated. Furthermore, the reviewer did not see too much mechanical insights of STIC2 to explain its role in the cpSRP pathway beyond the Bedard's paper.

Answer:

Since Reviewer #2 and Reviewer #3 express concerns about the novelty of our manuscript, we would like to refer here to our responses to points 1 and 2 of Reviewer #3 in order not to list the arguments redundantly.

Based on the current data, it's very hard to reach the conclusion that STIC2 is a specific factor which only delivers the co-translational substrates in the cpSRP pathway.

Answer:

We do not claim in the manuscript that STIC2 is exclusively involved in cotranslational sorting. As stated in the original manuscript's discussion and discussed in the revised version, we believe that STIC2 may also have roles in posttranslational sorting, as suggested by the study of Bedard et al., 2017 (lines 560-565).

Here are some **major concerns**:

1. *The data supporting STIC2 association with ribosomes is rather weak. It looks to me that the co-migration of STIC2 with ribosomes on the size exclusion chromatography is not convincing. The author argue that prepared ribosomes are largely translationally inactive or predominantly translate soluble proteins. It would be nice to see the better comigration of STIC2 and translating ribosomes on the thylakoid membrane if the membrane fraction is isolated for immunoblot.*

Answer:

In response to the reviewer's concerns, I would like to emphasize that the mass spectrometry and immunoblot data using affinity-purified ribosomes translating D1 (Figure 2C) clearly demonstrate an interaction between STIC2 and D1-RNCs. However, we agree that the data showing comigration of STIC2 with ribosomes using a stromal extract could be more robust.

To address this, we conducted new experiments. To maintain a high number of translating ribosomes, we now used an extract from freshly frozen leaf material. Sucrose density centrifugation assays now clearly demonstrate that a pool of STIC2 comigrates with ribosomes. These new findings replace the previous Figure 3A and are now presented as Figure 4A.

Additionally, we have included new data in the revised manuscript showing a translational defect in the *stic2* mutant (Figure 4C), which further supports a link between STIC2 and ribosomes.

2. *In the pull-down assay, STIC2 interacts much stronger with alb3 than with alb4. However, opposite conclusion was obtained from the ITC experiment. This discrepancy needs to be explained or tested by other experiments.*

Answer:

The pull-down assays shown in Figure 4B were replaced with new assays presented in the new Figure 6. These assays utilize the C-terminal regions of Alb3/Alb4 and STIC2 along with new site directed mutagenesis constructs of the interaction partners to characterize the binding interface. The updated experiments demonstrate a clear interaction of STIC2 with the C-terminal regions of both, Alb3 and Alb4. The new experiments were conducted to quantify the interaction between STIC2 and Alb3C variants as well as STIC2 and Alb4C variants. A direct comparison of the pull-down efficiency between Alb3C and Alb4C is difficult, because we applied different buffer conditions in the pull-downs using either Alb3C or Alb4C to minimize background that we especially observed using Alb3C, which tended to adhere to the control beads (please see also my reply to the minor point of Reviewer #2 below).

3. *It's difficult to understand that stic2 knockout mutant has no phenotype under any condition, given its role in D1 protein biogenesis. There are a few factors, like LPA1 or PAM68 have been reported to be involved in D1 protein assembly into the photosystem complex. Most of the mutants lacking these factors display severe phenotype of either growth or photosynthesis. Based on the proposed model, I expect to see some defects of the stic2 knockout mutant. The synergistic effect of STIC2 and cpSRP54 is obvious but also raises the question of how much percent of the nascent chains truly goes through STIC2. Does STIC2 interact with cpSRP54, or do they work independently? It's possible that STIC2 only mediates a very small number of D1 protein insertion and cpSRP54 plays more predominant roles. Can STIC2 overexpression rescue the phenotype of the ffc mutant at some level? In any way, the mechanism of STIC2 is not well clarified and this point should be the highlight of this study.*

Answer:

Firstly, I would like to mention that a previous study demonstrated that the *stic2* knockout mutant is characterized by an altered chloroplast ultrastructure, exhibiting features such as swollen chloroplasts, an increased number of plastoglobules and less tightly packed thylakoid lamellae (Bedard et al., 2017).

To get further insights into the phenotype of the *stic2* mutant we quantified the level of D1 and the other reaction center core subunits of PSII (D2) and PSI (PsaA, PsaB). Our new data show that the level of these proteins is slightly reduced compared to WT (see revised Figure 3B). This reduction, although subtle, indicate that the *stic2* mutant is impaired in thylakoid biogenesis. Moreover, new experiments to analyze the ratio of soluble and membrane-associated ribosomal footprints demonstrate that *stic2* accumulates soluble footprints, which indicates a role of STIC2 in translation (new Figure 4C). Furthermore, the ribosomal footprint analysis reveals that, unlike cpSRP54,

STIC2 does not significantly contribute to tethering of the ribosome to the thylakoid membrane. In conclusion, while the *stic2* plants do not exhibit a visual phenotype, there are distinct changes on a molecular level. These changes support a role of STIC2 in cotranslational sorting and suggest that its molecular function differs from cpSRP54.

We also investigated whether STIC2 interacts directly with cpSRP54 or its receptor, cpFtsY. However, our data do not support any direct interaction between STIC2 and cpSRP54 or cpFtsY. In response to the reviewer's question regarding the potential for STIC2 to compensate for cpSRP54 deficiency, we generated lines overexpressing STIC2 in the *ffc* mutant background. Preliminary analyses indicate that overexpression of STIC2 does not lead to complementation of the phenotype of the *ffc* mutant. Additionally, we did not observe upregulation of endogenous STIC2 in the *ffc* background or of cpSRP54 in the *stic2* mutant. Consistently, our data argue for distinct functional roles of STIC2 and cpSRP54. Because of our negative data regarding the interaction and complementation assays, we decided not to add these data to the manuscript.

Minor point:

- *The high background of the pulldown experiment shown in Fig S4 need to be addressed. There are still quite lots of protein bound non-specifically in the negative control. It might be worth to swap the bait and prey to repeat the experiment.*

Answer:

We reduced the background of our pulldown experiments and show all pulldowns in the new Figure 6 (please see also my reply to point 2 of Reviewer #2).

Referee #3:

In this manuscript, Stolle et al. report on the use of affinity-purification-based label-free quantitative proteomics, together with cell biology, biochemical and biophysical analyses and photosynthetic parameter measurements, to identify a D1 RNC interactome and then to point to a role of the STIC2 protein in D1 targeting and biogenesis. The manuscript includes a wealth of high-quality data that is nicely presented and so is very informative. This important study will be of considerable interest to those in the relevant intersecting fields of organelle biology, protein biogenesis, and photosynthesis. While the D1 RNC interactome is a significant dataset in its own right, the manuscript would be considerably strengthened if additional data could be added that provide significant new insights into how the STIC2 protein exerts its molecular function (i.e. mechanism) in D1 targeting and PSII biogenesis. Some specific comments are below:

1. As the authors pointed out in the Discussion, a direct interaction of STIC2 with both thylakoid membrane proteins, Alb3 and Alb4, had been reported before (Figure 6 in Bedard et al., 2017). In vitro pulldown experiments were used in both studies (e.g. Alb3 aa 350-462 and aa 361-462 were used here and in Bedard et al., 2017, respectively; Alb4 aa 334-499 and aa 345-498 were used here and in Bedard et al., 2017, respectively), and the same observations and conclusions were made (i.e. the conserved C-terminal motif III of Alb3 and Alb4 interacts with STIC2). It would greatly strengthen this manuscript if the data here could be extended to take it beyond what was previously reported.

Answer:

The first version of the manuscript already extends beyond the findings of Bedard et al., 2017 by providing a more comprehensive analysis of the interactions between STIC2 and the thylakoid membrane proteins, Alb3 and Alb4. Unlike Bedard et al., our study systematically screened the full-length Alb3 and Alb4 proteins to identify STIC2 binding motifs (Pepspot analysis, Figure 5A, Appendix Figure S5). This approach shows a direct binding between motif III and STIC2, excluding the possibility that the G397 mutation on Alb4 affects the STIC2 interaction indirectly by misfolding of Alb4. Additionally, we quantitatively analyzed the interactions between STIC2 and the C-terminal regions of Alb3C and Alb4C (ITC data). These data point to Alb4 as the primary interaction partner. Nevertheless, we acknowledge the Reviewer's suggestion to further extend the interactions studies to enhance the manuscript's impact. Therefore, we performed additional experiments to pinpoint the docking site of motif III in STIC2, which has not been described yet. We chose a bioinformatic approach using AlphaFold to generate a structural model of the Alb4C/STIC2 complex. The model identified motif III and the β -sheet region of STIC2 as the primary binding interface with a high level of confidence. To validate this model, we conducted pull-down experiments using site directed mutagenesis constructs of Alb4C and STIC2. The results, now presented in the new Figure 6, provide substantial new insight into the interaction of STIC2 with Alb4.

*2. The same genetic analysis (i.e. the single and double mutants of *stic2* and *ffc1-2=cpsrp54*) was conducted here*

and by Bedard et al., 2017. Although different methods (e.g. Fv/Fm measurements, immunoblotting, and pulse-chase assay) were used here to compare the phenotype between the different genetic backgrounds, this new analysis does not provide much additional insight into how STIC2 exerts its functions in D1 synthesis or STIC2's relationship with the cpSRP54 pathway. Again, it would add considerable value to the manuscript if significant new insight could be added here.

Answer:

In Bedard et al., 2017, the analysis of the *stic2 srp54* double mutant was limited to a description of the visual phenotype without an analysis on a molecular level. Bedard et al. suggested a role for STIC2 “in thylakoid protein targeting, potentially for a specific subset of thylakoidal proteins”, based on the observed genetic interaction with SRP pathway components (documented by the visual phenotype of the mutants) and the physical interaction between STIC2 and the Alb3/4 insertase. However, they did not provide data demonstrating that the double mutant is affected in thylakoid membrane biogenesis or elucidate how this process might be disrupted. We find the reviewer’s comment that our “new analysis does not provide much additional insight into how STIC2 exerts its functions in D1 synthesis” somewhat surprising. We like to emphasize that it has not been demonstrated yet, that STIC2 is involved in D1 synthesis. While the genetic interaction between *stic2* and *cpSRP54* observed by Bedard et al. is consistent with such a function, it could also be due to other physiological defects.

Our study offers significant new insights into the molecular functions of STIC2. We demonstrate that STIC2 is associated with ribosomes and show that mutants lacking both STIC2 and cpSRP54 suffer from a severe impairment in D1 synthesis. During the revision process we expanded our findings with additional data and refined analysis. We performed a quantitative analysis of our data and extended our immunoblot analysis to include more reaction center subunits of PSII and PSI (see updated Figure 3B, previously Figure 5B). Additionally, we now provide even stronger evidence of STIC2's binding to ribosomes (new Figure 4A, please see also our response to Reviewer #2 for more details). We also present new data from a ribosomal footprint analysis of the mutants (new Figure 4C), which indicates that the STIC2 mutation leads to a translational defect. In conclusion, our analysis offers substantial new insights into the function of STIC2 by demonstrating its involvement in cotranslational sorting of D1 and potentially other key subunits of the PSI and PSII.

3. One addition that would considerably add to the novelty/conceptual advance here would be the identification and characterization of the binding site between STIC2 and the D1 RNC, and how STIC2 is recruited the D1 RNC.

Answer:

We agree with the reviewer that elucidating the molecular details of the dynamic interaction between STIC2 and the D1 RNC would indeed be very interesting and valuable. However, investigating these aspects would require an extensive series of experiments that go far beyond the scope of this study. We think that these aspects could be addressed in a follow-up project. The detailed data to elucidate this mechanism would likely be significant enough for a separate high-quality publication.

4. Fold-change of proteins with significant quantitative differences (i.e. enriched) needs to be presented in the tables and supplemental datasets.

Answer:

We added this information.

5. Figure 5C. A full gel of the pulse-labelling assay is required to see whether the observed effect is D1 specific or is due to a potential methionine uptake issue in the double mutant. Quantitative analysis of these data (C and D), with replicates and statistical analysis, is also needed.

Answer:

Regarding the quantification and additional labeling signals, please see my comment to Reviewer #1.

6. Line 456. Alb4 also contains motif II according to Trösch et al. (2015b).

Answer:

We have largely revised this section of the discussion. However, to concentrate on the molecular function of STIC2 and to streamline the discussion we have chosen to remove the part concerning motif II.

7. Lines 470-474. Here, it would be helpful if the authors would comment on the hypotheses proposed by Bedard et al. (2017).

Answer:

We rephrased this section of the discussion.

Dear Danja,

We have now received re-review reports from two referees. As you will see, you have addressed their concerns satisfactorily. You will see, though, that Referee #1 raises a number of observations and queries; these should either be directly corrected (please also consider Referee #2's point about data presentation), or dealt with in the discussion section. Before I can finally accept the manuscript, there are some remaining editorial points which need to be addressed. In this regard would you please:

- include acknowledgement of funding from the Max Planck Society and DFG (ZO 302/5-1) in our online submission platform,
- rename the conflict of interest statement the "Disclosure and competing interests statement",
- correct the callout for Table S1 to Appendix Table S1,
- upload dataset legends separate sheets in each Excel file,
- correct nomenclature to Appendix S1-S6 and Appendix Table S1-S2 in figure/table legends, citing references as 10 authors + et al. instead of 6 authors + et al.,
- state in the legend if the dot-blot image has been re-used in Figure 5A & Appendix Fig S5 A and B,
- provide a specific URL for dataset PXD042896, and
- provide exact p values in the legend of figure 3a.

EMBO Press is an editorially independent publishing platform for the development of EMBO scientific publications.

Best wishes,

William

William Teale, PhD
Editor
The EMBO Journal
w.teale@embojournal.org

We realize that it is difficult to revise to a specific deadline. In the interest of protecting the conceptual advance provided by the work, we recommend a revision within 3 months (7th Oct 2024). Please discuss the revision progress ahead of this time with the

editor if you require more time to complete the revisions. Use the link below to submit your revision:

Referee #1:

The authors have thoroughly addressed my critical points. Most importantly, they have included quantifications of the immunoblot and pulse labeling data as well as site-directed mutagenesis data that both strengthen the manuscript. I still have a couple of minor points (most of them emerging with the new/revised data) that need to be addressed:

- Line 158: There is a discrepancy in the number of total proteins in the text (259) and in Figure 2a (251)
- Line 163: The number of proteins in the two datasets do not match the numbers indicated in the text (122 in EV1 and 136 in EV2)
- Line 229: ...in D1-RNCs with medium long chains... (not longer chains; or be more specific regarding the length)
- Line 238: TIG1 is not specifically recruited to D1 translating ribosomes, it is equally present in the TST-uS2c control
- Line 247: better say: members of the HSP100, HSP90 and HSP70 molecular chaperone systems
- Line 248: HSP90-5, not HS905
- Line 303: The authors state: 'Pulse-chase experiments showed no pronounced differences in D1 degradation between the mutants and WT'. This is not true, the fraction of labeled D1 after 5 h chase is 30% in the WT and 15% in the double mutant. On what basis were these samples loaded?
- Line 308: The authors state in their response letter that the lack of the RC in the double mutant was not reproducible. This information needs to be given somewhere, otherwise the statement simply does not match with what is shown. I wonder if the RC is prone to degradation in the double mutant, which would also explain the reduced stability of D1 indicated by the chase experiment?
- Line 358: The increase in soluble footprints in the double mutant is not higher than in the ffc single mutant. I doubt that the drop in membrane footprints in the double mutant is significant.
- Line 364: but how can it be explained that in the absence of STIC2 the reduced membrane tethering observed in the absence of cpSRB43 is restored?!
- Line 429 (G111), line 433 (K115A), line 437 (E112A/K115A): the intensities shown in the immunoblots do not really match the quantification values. Ideally the authors would have loaded 100%, 50% and 25% of the STIC2 WT sample to make this clearer.
- Line 529: Instead of 'Chaperonin, heat shock proteins' better use 'Molecular chaperones'.

Referee #2:

The authors have clarified all my concerns and updated some data in the revised manuscript. I have only one minor suggestion. The new Figure 4C could be improved to show the ratio between soluble and membrane footprint by chart, as the author argued that the ratio is significantly changed in line 354-357.

Point-by-point response

Referee #1:

The authors have thoroughly addressed my critical points. Most importantly, they have included quantifications of the immunoblot and pulse labeling data as well as site-directed mutagenesis data that both strengthen the manuscript. I still have a couple of minor points (most of them emerging with the new/revised data) that need to be addressed:

- Line 158: There is a discrepancy in the number of total proteins in the text (259) and in Figure 2a (251)

Answer:

A total of 259 proteins were identified in the MS analysis, based on individual Psat numbers as indicated in the previous Dataset EV4. In the revised manuscript, we now refer to this dataset as Dataset EV1: "A total of 259 proteins were identified (Dataset EV1)". As explained in the text, 251 of these proteins were either enriched in the D1 samples or equally present in both the D1 and uS2 samples. The remaining 8 proteins, while not explicitly mentioned, do not fall into these categories and are implied to be specific to the uS2 samples. As these proteins are not relevant to the main focus of the manuscript, we chose not to explicitly mention them.

- Line 163: The number of proteins in the two datasets do not match the numbers indicated in the text (122 in EV1 and 136 in EV2)

Answer:

In Dataset EV2 (now EV3), two Psat numbers were assigned to one Uniprot ID or *A. thaliana* accession on five occasions. As we counted the Psat accessions, this summed up to 141 as mentioned in the text. In Dataset EV1 (now EV2), we counted the Psat accessions excluding the IDs marked with an asterisk, which resulted in 110 hits. In the revised version, we provide more detailed information in the legends of the datasets to clarify this discrepancy.

- Line 229: ...in D1-RNCs with medium long chains... (not longer chains; or be more specific regarding the length)

Answer:

We changed the text as suggested.

- Line 238: TIG1 is not specifically recruited to D1 translating ribosomes, it is equally present in the TST-uS2c control

Answer:

Yes, we apologize for the mistake. This has been corrected accordingly.

- Line 247: better say: members of the HSP100, HSP90 and HSP70 molecular chaperone systems

Answer:

Yes, we changed the text as suggested.

- Line 248: HSP90-5, not HS905

Answer:

We corrected this.

- Line 303: The authors state: 'Pulse-chase experiments showed no pronounced differences in D1 degradation between the mutants and WT'. This is not true, the fraction of labeled D1 after 5 h chase is 30% in the WT and 15% in the double mutant. On what basis were these samples loaded?

Answer:

To address this point, we added quantification data from a second degradation experiment to Appendix Figure S4A. As stated in the text, we do not find pronounced differences in D1 degradation.

Samples were loaded based on equivalence of leaf discs (see Materials and Methods). As explained in our initial response letter, it is not feasible to include a signal from an unrelated equally labeled protein as a control. Additionally, using a Coomassie-stained gel for loading controls between mutants is also not suitable due to the significant differences in protein patterns.

- Line 308: The authors state in their response letter that the lack of the RC in the double mutant was not reproducible. This information needs to be given somewhere, otherwise the statement simply does not match with what is shown. I wonder if the RC is prone to degradation in the double mutant, which would also explain the reduced stability of D1 indicated by the chase experiment?

Answer:

Yes, we agree that it is better to provide a more representative immunoblot showing the presence of the RC in the double mutant. In response, we replaced the previous blot with a new one that clearly demonstrates the presence of the RC in the double mutant. Initially, we hesitated to use this blot because the separation of the various complexes is less distinct compared to the previous immunoblot. However, to address the concern and ensure clarity, we now included this blot.

- Line 358: The increase in soluble footprints in the double mutant is not higher than in the ffc single mutant. I doubt that the drop in membrane footprints in the double mutant is significant.
- Line 364: but how can it be explained that in the absence of STIC2 the reduced membrane tethering observed in the absence of cpSRB43 is restored?!

Answer:

It is true that the phenotype of the double mutant cannot be simply explained by the combination of the phenotypes of the single mutants. This is one reason why we explicitly state in the text that the results leave room for speculation. One possibility is that the double mutant exhibits an alternative way of tethering translating ribosomes to the membrane that becomes accessible in the absence of both STIC2 and cpSRP54. However, this binding is apparently not productive for ongoing translation and insertion. This could explain the only slight decrease of membrane-bound footprints and the increased amount of soluble footprints in the double mutant.

To address this critical point, we added the following sentence to the discussion: “Unexpectedly, the double mutant shows only a slight decrease in membrane-bound footprints. This may imply that in absence of both STIC2 and cpSRP54, ribosome binding occurs through an alternative mechanism, that however, does not support efficient cotranslational insertion.”

- Line 429 (G111), line 433 (K115A), line 437 (E112A/K115A): the intensities shown in the immunoblots do not really match the quantification values. Ideally the authors would have loaded 100%, 50% and 25% of the STIC2 WT sample to make this clearer.

Answer:

We selected the most representative uncut immunoblots and believe that the data adequately reflect the quantification results. For the G111 sample, the standard deviation of the quantification is relatively high, which accounts for the slight discrepancy between the blot and the quantification values.

- Line 529: Instead of 'Chaperonin, heat shock proteins' better use 'Molecular chaperones'.

Answer:

We changed that accordingly.

Referee #2:

The authors have clarified all my concerns and updated some data in the revised manuscript. I have only one minor suggestion. The new Figure 4C could be improved to show the ratio between soluble and membrane footprint by chart, as the author argued that the ratio is significantly changed in line 354-357.

Answer:

During the first revision of the manuscript, we have carefully considered how to best present the new data. We decided not to present the footprint ratios because we believe that presenting the soluble and membrane-bound footprints separately provides the clearest and most succinct representation of the results. Showing the ratios might obscure the fact that the membrane-bound footprints are not reduced in the STIC2 mutant and only slightly reduced in the double mutant. Therefore, we would like to retain our current method of data presentation.

Editor:

- include acknowledgement of funding from the Max Planck Society and DFG (ZO 302/5-1) in our online submission platform,

Answer:

We included this information.

- rename the conflict of interest statement the "Disclosure and competing interests statement",

Answer:

We renamed the statement.

- correct the callout for Table S1 to Appendix Table S1,

Answer:

We changed that.

- upload dataset legends separate sheets in each Excel file,

Answer:

We changed the files accordingly.

- correct nomenclature to Appendix S1-S6 and Appendix Table S1-S2 in figure/table legends, citing references as 10 authors + et al. instead of 6 authors + et al.,

Answer:

We corrected that and changed the citations.

- state in the legend if the dot-blot image has been re-used in Figure 5A & Appendix Fig S5 A and B,

Answer:

We added the following statement to the legend of Appendix Fig S5: "The peptide arrays shown in the right panels of this figure and in Figure 5A are identical."

- provide a specific URL for dataset PXD042896, and

Answer:

To make the data accessible to readers, we need to inform the PRIDE team (EMBL-EBI) that the publication of the corresponding manuscript is online. Subsequently, the data will become available via ProteomeXchange with identifier PXD042896. Unfortunately, the coauthor who has the login ID for the PRIDE database is on vacation. We will inform PRIDE as soon as possible. In the revised version we adapted the text according to the instructions of PRIDE for published manuscripts.

- provide exact p values in the legend of figure 3a.

Answer:

We added this information.

Dear Danja,

I am pleased to inform you that your manuscript has been accepted for publication in the EMBO Journal.

Congratulations! I am really happy to have this article in The EMBO Journal.

Best wishes,

William

William Teale, PhD
Editor
The EMBO Journal
w.teale@embojournal.org
